# The PP2A-like phosphatase Ppg1 mediates assembly of the Far complex to balance gluconeogenic outputs and enables adaptation to glucose depletion

Shreyas Niphadkar[1,2], Lavanya Karinje[1], Sunil Laxman[1]*

1 Institute for Stem Cell Science and Regenerative Medicine (DBT-inStem) Bangalore, India, 2 Manipal Academy of Higher Education, Manipal, Karnataka, India

* sunil@instem.res.in

**Data Availability Statement:** All data used are provided in the manuscript and in the supplemental information without restriction. Proteomics data are also deposited in the PRIDE public database,

## Abstract

To sustain growth in changing nutrient conditions, cells reorganize outputs of metabolic networks and appropriately reallocate resources. Signaling by reversible protein phosphorylation can control such metabolic adaptations. In contrast to kinases, the functions of phosphatases that enable metabolic adaptation as glucose depletes are poorly studied. Using a *Saccharomyces cerevisiae* deletion screen, we identified the PP2A-like phosphatase Ppg1 as required for appropriate carbon allocations towards gluconeogenic outputs—trehalose, glycogen, UDP-glucose, UDP-GlcNAc—after glucose depletion. This Ppg1 function is mediated via regulation of the assembly of the Far complex—a multi-subunit complex that tethers to the ER and mitochondrial outer membranes forming localized signaling hubs. The Far complex assembly is Ppg1 catalytic activity-dependent. Ppg1 regulates the phosphorylation status of multiple ser/thr residues on Far11 to enable the proper assembly of the Far complex. The assembled Far complex is required to maintain gluconeogenic outputs after glucose depletion. Glucose in turn regulates Far complex amounts. This Ppg1-mediated Far complex assembly, and Ppg1-Far complex dependent control of gluconeogenic outputs enables adaptive growth under glucose depletion. Our study illustrates how protein dephosphorylation is required for the assembly of a multi-protein scaffold present in localized cytosolic pools, to thereby alter gluconeogenic flux and enable cells to metabolically adapt to nutrient fluctuations.

## Author summary

Many organisms live in changing nutrient environments. To grow effectively, cells continuously adapt to changing nutrient conditions (such as glucose limitation) and reorganize their metabolic outputs. For this adaptation, cells employ 'signaling systems' that respond to nutrient conditions and regulate metabolism. Protein phosphatases in eukaryotic cells are part of one such signaling system which regulates metabolic outputs. They function by dephosphorylating target proteins to change their activity or function. In this study, we

and the data are available via ProteomeXchange with identifiers PXD049793 and PXD049989. The appropriate supplemental tables for raw data are cited in the methods section and/or text.

**Funding:** The Wellcome Trust DBT India Alliance (India Alliance):SL, IA/S/21/2/505922. The funders had no role in study design, data collection and analysis, decision to publish, or preparation of the manuscript.

**Competing interests:** The authors have declared that no competing interests exist.

used budding yeast (*Saccharomyces cerevisiae*) as a model to study signaling-based adaptation to glucose limitation. Using a screen, we identified a protein phosphatase Ppg1, a member of PP2A family, as a regulator of metabolic outputs after glucose depletion. Surprisingly, we found that Ppg1 carried out this function by regulating the assembly of a large protein complex, Far complex. Ppg1 phosphatase dephosphorylated a key protein of this complex thereby allowing the complex to assemble. This assembled Far complex controlled the metabolic reorganization. This study therefore finds a novel mechanism through which cells fine-tune metabolism to thrive after glucose depletion, via the formation of a large protein complex in cells. This study deepens our understanding of signaling responses in changing environments required for appropriate metabolic adaptation.

## Introduction

Cells continuously adapt to fluctuating nutrient conditions by rewiring their metabolism to efficiently utilize available nutrients. Cells utilize multiple signaling systems for this metabolic rewiring, thereby facilitating adaptation to these environments [1–5]. Therefore, understanding how signaling processes respond to changing conditions and regulate metabolism is of obvious interest. We have a growing understanding of signaling mechanisms that are active in specific nutrient environments, but our understanding of their roles in changing nutrient environments remains incomplete. However, this understanding is important to comprehend the basic principles of metabolic adaptation, with useful applications in industrial settings.

*Saccharomyces cerevisiae* is a remarkable model system using which several molecular mechanisms of metabolic adaptation, and conserved signaling systems that regulate these have been discovered [4,6]. Particularly, this system has been instrumental in advancing our understanding of mechanisms of how cells adapt as glucose levels change. In glucose-replete conditions, *S. cerevisiae* cells preferentially utilize glucose as a carbon source and repress the utilization of other carbon sources, as well as mitochondrial respiration [7–9]. This includes repression of metabolic processes utilizing alternative carbon sources such as gluconeogenesis, glyoxylate cycle, TCA cycle, etc [10]. When in glucose-replete conditions, *S. cerevisiae* cells adapt over the course of growth as they rapidly consume and deplete glucose [11]. As glucose depletes, *S. cerevisiae* cells undergo a diauxic shift where cells reorganize their metabolic networks to utilize other available carbon sources [12–14]. In the post-diauxic phase, cells increase mitochondrial respiration and initiate gluconeogenesis to produce biomass precursors for growth. The outputs of gluconeogenesis include trehalose, glycogen, UDP-glucose, UDP N-acetyl glucosamine and others, all of which have essential roles in growth and adaptation [15,16]. While the roles of several signaling proteins in regulating carbon metabolism in high or low glucose respectively have been studied [17,18], our understanding of signaling mechanisms that regulate metabolic adaptations when glucose depletes remains incomplete.

Signaling mechanisms involve signal relays regulated by post-translational modifications (PTMs), particularly reversible protein phosphorylation, and can regulate metabolism [19–22]. In such signal relays, there typically is reciprocal regulation by writer-eraser systems, such as kinases and phosphatases [23]. In the context of glucose responses studied in yeast, our current understanding is kinase centric. In glucose-replete conditions, we have a growing understanding of regulation mediated by protein kinases such as protein kinase A (PKA) and TOR kinase (target of rapamycin), which activate growth programs in glucose-replete conditions. Specifically, these kinases activate transcription programs that control carbon metabolism, ribosome biogenesis, stress response, etc [4,24–29]. Upon glucose depletion, the Snf1 kinase

(AMP-activated protein kinase) is activated and drives the utilization of other carbon sources via gluconeogenesis, glyoxylate cycle, and TCA cycle [30–32]. Additionally, glucose starvation activates the Rim15 kinase, which controls storage carbohydrate metabolism and stress responses [33–35]. Given the importance of regulating gluconeogenic outputs as glucose depletes, which involves extensive phosphorylation-based regulation, control by protein dephosphorylation would be anticipated for such metabolic adaptation. However, the known roles of phosphatases in carbon metabolism are limited to a few examples. The PP1 phosphatase dephosphorylates and inactivates the Snf1 kinase to enable glucose-mediated catabolite repression in glucose-replete conditions [36,37]. PP1 also dephosphorylates a component of eisosomes in glucose-depleted conditions to regulate nutrient transporter availability at the plasma membrane, enabling cells to recover from glucose starvation [38]. Upon glucose starvation, PP2A phosphatases dephosphorylate stress-responsive transcription factors to transcriptionally regulate enzymes that increase the synthesis of storage carbohydrates [34,39–41]. Given the number of phosphatases in eukaryotes including yeast [40], it is plausible that other protein phosphatases play other roles in metabolic adaptations to changing glucose environments.

Here, using a phosphatase knockout screen, we identify a role for the PP2A-like phosphatase Ppg1, in regulating metabolic adaptations in the post-diauxic phase. Using *ppg1Δ* and Ppg1 catalytically-inactive cells, we establish the role of Ppg1 in controlling gluconeogenic flux and carbon allocations post glucose depletion. We identify that Ppg1 carries out this function independent of transcriptionally regulating storage carbohydrate synthesis. Instead this Ppg1 function is via controlling the assembly of a multiprotein scaffolding complex—the Far complex. The Ppg1 and Far complex-mediated regulation of gluconeogenic outputs are critical for cells to adapt to glucose depletion in competitive growth environments where glucose gets depleted rapidly. Collectively, we identify a mechanism where cells use protein dephosphorylation to dynamically assemble a scaffolding complex on organelles as assembly hubs, through which cells can tune carbon allocations in order to adapt to fluctuating nutrient environments.

## Results

### A phosphatase deletion screen identifies Ppg1 as a regulator of storage carbohydrate metabolism

As glucose depletes, Crabtree-positive cells such as *S. cerevisiae* gradually switch from glycolytic fermentation to gluconeogenesis and the utilization of other carbon sources (Fig 1A). Our objective was to identify protein phosphatases that regulate this metabolic adaptation (Fig 1A). To address this, we generated a *S. cerevisiae* phosphatase deletion library consisting of individual deletions of 38 non-essential phosphatases (S1A Fig), and carried out a screen to identify phosphatases that regulate carbon metabolism post glucose depletion. For this screen, wild-type and phosphatase deletion mutant cells were cultured in a glucose-replete medium (YPD), and grown post glucose depletion (post-diauxic phase). For the screen output, trehalose accumulation was assessed from these mutants after 24hrs. At this 24hrs time point, no glucose was detected in the medium, confirming that cells are in a post-diauxic phase. Trehalose synthesis increases in the post-diauxic phase and is a reliable readout of a gluconeogenic state [16,42] (Fig 1B). We initially estimated how much trehalose amounts increased in the post diauxic phase. Trehalose amounts increased over 10-fold in the post-diauxic phase after 24 hours of growth starting in 2% glucose, compared to cells after 4 hours of growth in the same condition (S1B Fig). Using this approach, we identified phosphatase mutants with increased or decreased trehalose accumulation (Figs 1C and S1A). A notable 'hit' was the PP2A-like phosphatase Ppg1, where the loss of Ppg1 results in a significant increase in trehalose accumulation in the

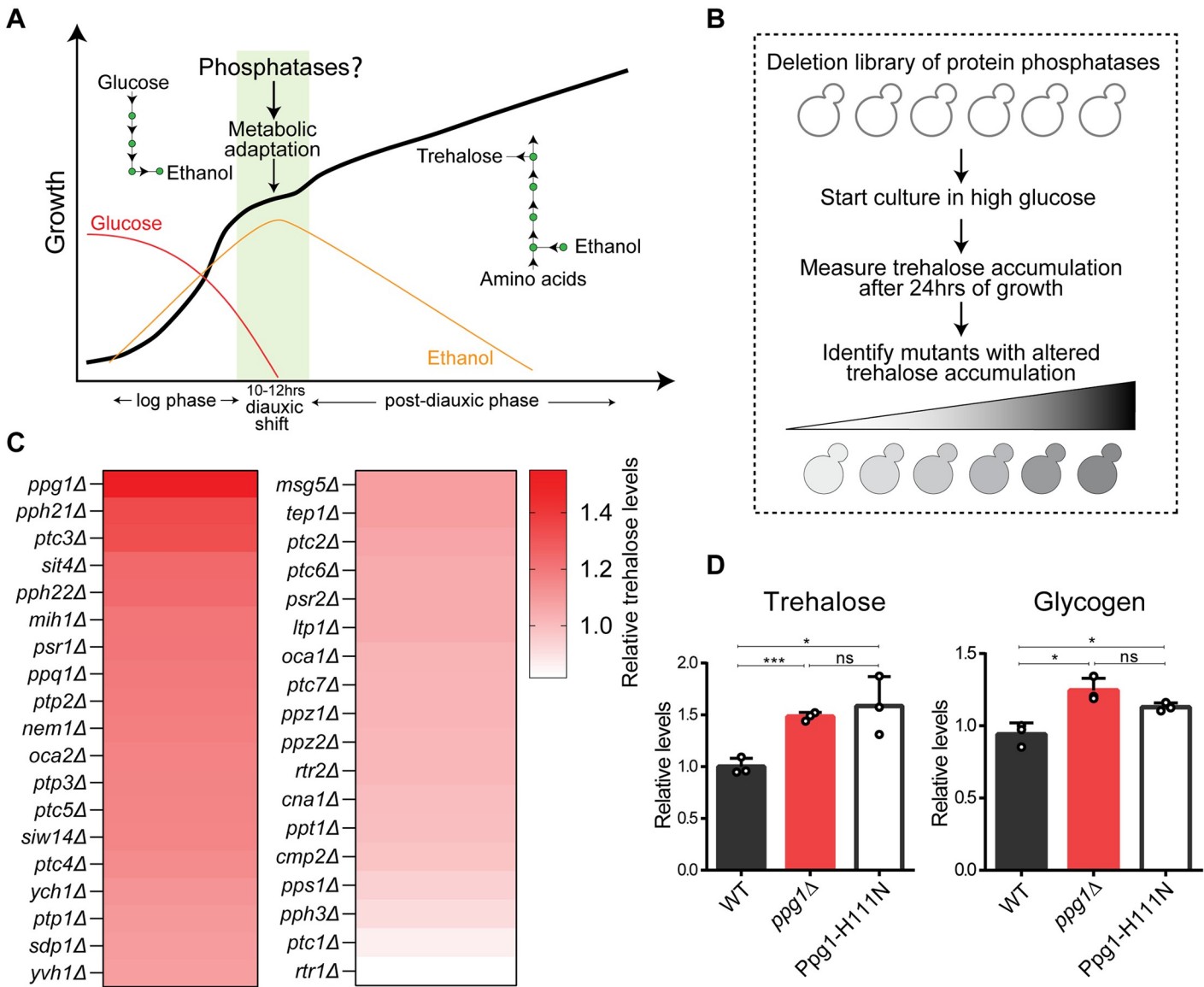

**Fig 1. A phosphatase deletion screen identifies Ppg1 as a regulator of storage carbohydrate metabolism.** A) Schematic depicting growth kinetics of *S. cerevisiae* cells in glucose-replete conditions and varying metabolic states during different phases of growth. *S cerevisiae* cells rewire their metabolism after diauxic adaptation. B) Schematic describing the screen designed to identify protein phosphatases that regulate post-diauxic carbon metabolism. For the screen, phosphatase knockouts were started in a glucose-replete medium, and trehalose accumulation was measured after 24hrs of growth. Also, see S1A Fig for the table of phosphatase mutants used in this study. C) Identifying phosphatase knockouts with altered trehalose amounts. The heat map shows relative trehalose amounts in phosphatase knockouts compared to wild-type cells. Trehalose accumulation was measured after 24hrs of growth in YPD medium. The mean trehalose levels were from 2 biological replicates. Also, see S1A Fig for steady-state trehalose amounts. D) Effect of loss of phosphatase activity of Ppg1 on the accumulation of storage carbohydrates in post-diauxic phase. A catalytically inactive mutant of Ppg1-Ppg1$^{H111N}$ was generated. WT, *ppg1Δ*, and Ppg1$^{H111N}$ cells were grown in a glucose-replete medium and trehalose and glycogen were measured after 24hrs. Data represented as a mean ± SD (n = 3). *P < 0.05, **P < 0.01, and ***P< 0.001; n.s., non-significant difference, calculated using unpaired Student's *t* tests. Also see S1B, S1D and S1E Fig.

post-diauxic phase (Fig 1C). Compared to the ~10 fold increase in trehalose (as shown in S1B Fig), the further increase of 1.5-fold in trehalose accumulation in the post-diauxic phase in cells lacking Ppg1 (Fig 1D) is substantial. This increased trehalose accumulation in post-diauxic *ppg1Δ* cells suggested that Ppg1 might balance carbon metabolism upon glucose depletion. We therefore focused on deciphering roles of Ppg1 in these processes. Currently, there is

limited evidence for Ppg1 in regulating glycogen metabolism [43]; and its roles or mechanisms in regulating metabolic adaptation are unknown.

While little is known about any Ppg1 functions in maintaining metabolic homeostasis, a previous study had identified a role for Ppg1 in preventing mitophagy, by dephosphorylating the autophagy-regulating protein Atg32 [44]. Although the conditions in this study do not induce mitophagy, we asked if these distinct roles were connected, using Atg32 mutants. However, a loss in this mitophagy regulator had no impact on trehalose accumulation (S1C Fig), suggesting that this function of Ppg1 in controlling trehalose metabolism is independent of its role in mitophagy. We next assessed if both glycogen and trehalose (which are synthesized from a common node of glucose-6-phosphate produced by gluconeogenesis) were regulated by Ppg1. The loss of Ppg1 phosphatase showed increased glycogen accumulation in post-diauxic phase (Fig 1D), similar to the accumulation of trehalose. In order to determine if this function of Ppg1 in storage carbohydrate metabolism depends on its phosphatase activity, we generated a catalytic-dead single point mutant of Ppg1 –Ppg1^H111N at the native Ppg1 chromosomal locus [45], and measured levels of trehalose and glycogen in these cells. Similar to *ppg1Δ* cells, Ppg1^H111N cells showed an increase in amounts of trehalose and glycogen in post-diauxic phase, confirming that the phosphatase activity of Ppg1 is required to control storage carbohydrate metabolism (Fig 1D). As a further control, we also confirmed that this point mutation in the catalytic site did not affect steady-state levels of Ppg1 protein and the addition of the FLAG epitope does not affect Ppg1 function (S1D and S1E Fig).

Collectively these data establish that the loss of Ppg1 leads to an accumulation of storage carbohydrates in the post-diauxic phase, and this depends on Ppg1 phosphatase activity.

## Ppg1 controls gluconeogenic flux and outputs in the post-diauxic phase

Since the loss of Ppg1 increased storage carbohydrates, we asked if other gluconeogenic outputs are altered in these cells. To address this, we used quantitative, targeted LC-MS/MS to identify changes in carbon allocation to other metabolic arms. We first assessed relative amounts of these metabolites (steady-state) from WT and *ppg1Δ* cells after 24hrs of growth in YPD. Notably, steady-state amounts of glucose-6-phosphate (G6P), fructose-6-phosphate (F6P), fructose-1,6-bisphosphate (F16BP) (which would all come from gluconeogenesis post glucose depletion) significantly increased in *ppg1Δ* cells. The levels of UDP-Glucose—a precursor of storage carbohydrates and cell wall increased in *ppg1Δ* cells. UDP-GlcNAc (a precursor of the chitin component of cell wall) was also higher in *ppg1Δ* cells (Fig 2A). Upon glucose depletion, amino acids can provide carbon backbones for gluconeogenesis [46]. We therefore also assessed the amounts of amino acids in these cells. The steady-state amounts of multiple free amino acids significantly decreased in *ppg1Δ* cells (Fig 2A). Collectively, these data suggest that *ppg1Δ* cells have imbalanced carbon metabolism, where amino acids might be consumed at an increased rate fueling gluconeogenesis, and carbon flux may be directed towards synthesis of storage carbohydrates and cell wall precursors. This was also observed in Ppg1^H111N cells, confirming the role of Ppg1 phosphatase activity in regulating carbon metabolism post glucose depletion (S2A Fig).

Since steady-state metabolite levels do not always indicate changes in flux, in order to unambiguously address if carbon flux was imbalanced in *ppg1Δ* cells, we compared carbon flux through these pathways in WT and *ppg1Δ* cells. Specifically, we used a pulse label of 1% $^{13}C_2$-acetate (which can directly enter gluconeogenesis) to cells in the post-diauxic phase, and estimated label incorporation into gluconeogenic intermediates, precursors of cell wall and storage carbohydrates. The relative $^{13}C$ label incorporation into gluconeogenic intermediates significantly increased, as did the label incorporation into trehalose, UDP-Glc, and

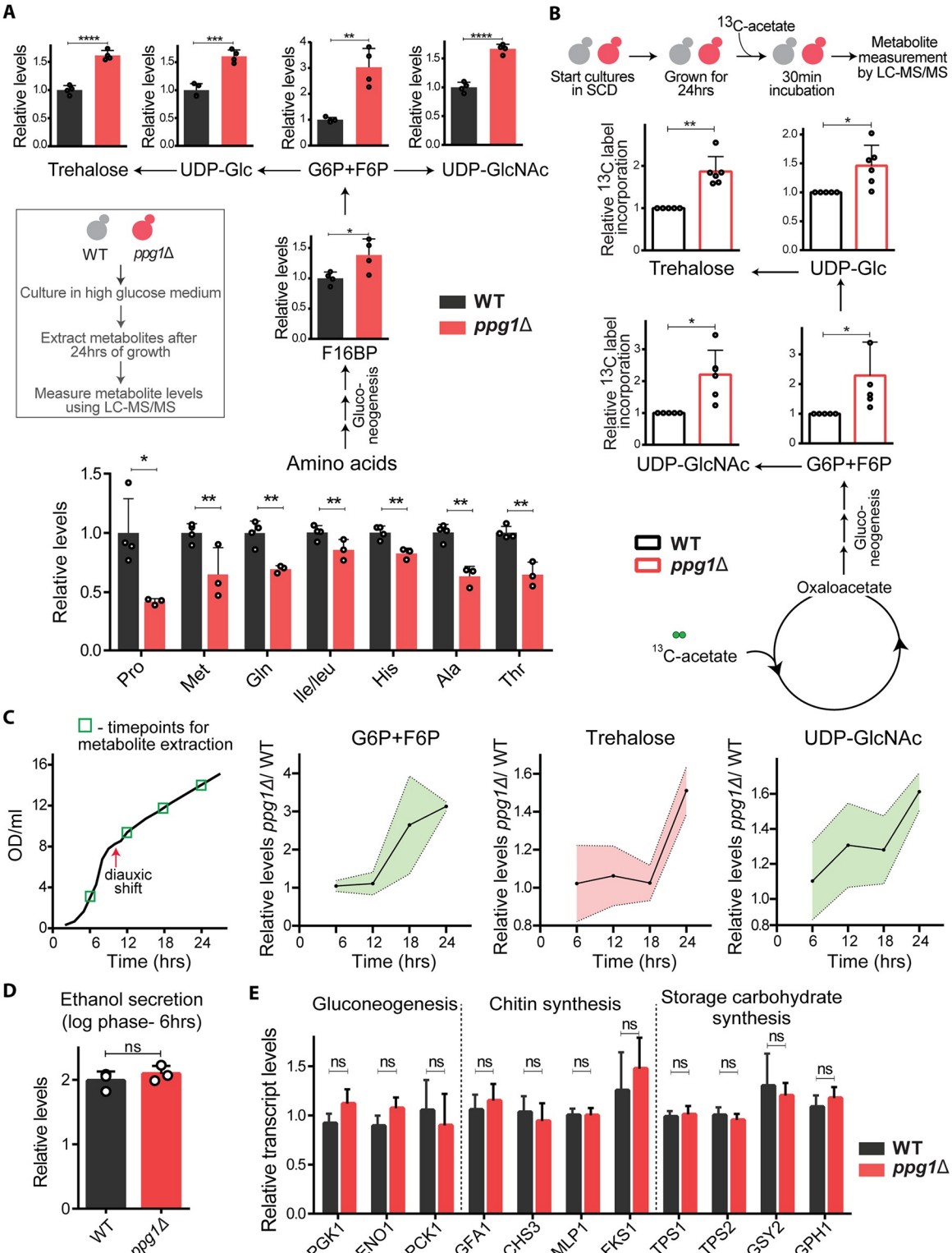

**Fig 2. Ppg1 controls gluconeogenic flux and outputs in the post-diauxic phase.** A) Relative steady-state amounts of specific gluconeogenic intermediates, precursors of cell wall and storage carbohydrates, and amino acids in WT and *ppg1Δ* cells after 24hrs of growth in YPD medium. Data represented as a mean ± SD (n = 3). G6P, glucose-6-phosphate; F6P, fructose-6-phosphate; UDP-Glc, uridine diphosphate glucose; UDP-GlcNAc, uridine diphosphate N-acetylglucosamine. *P < 0.05, **P < 0.01, and ***P < 0.001; n.s., non-significant difference, calculated using unpaired Student's *t* tests. Also see S2A Fig. B) Relative 13C label incorporation in

gluconeogenic outputs. WT and *ppg1Δ* cells were grown in SCD medium for 24hrs, cultures were spiked with $^{13}$C-acetate for 30mins, and $^{13}$C label incorporation in gluconeogenic outputs was measured. Data represented as a mean ± SD (n = 6). *P < 0.05, **P < 0.01, and ***P< 0.001; n.s., non-significant difference, calculated using paired Student's *t* tests. C) Relative steady-state amounts of specific gluconeogenic outputs from WT and *ppg1Δ* cells during the course of growth in YPD medium. WT and *ppg1Δ* cells were grown in YPD medium, and metabolite extraction was carried out at indicated time points. Data represented as a mean ± SD (n = 3). Also see S2B Fig for chitin levels, and S2C and S2D Fig, for growth in the presence of Congo red or Calcofluor white. D) Relative ethanol production by WT and *ppg1Δ* cells in high-glucose conditions. WT and *ppg1Δ* cells were grown in YPD medium for 6hrs and ethanol concentration in the media was measured. Data represented as a mean ± SD (n = 3). *P < 0.05, **P < 0.01, and ***P< 0.001; n.s., non-significant difference, calculated using unpaired Student's *t* tests. E) Effect of the loss of Ppg1 on transcript levels of gluconeogenesis, cell wall synthesis and storage carbohydrate synthesis genes. The transcript levels were measured after 24hrs of growth in YPD medium. Data represented as a mean ± SD (n = 3). *P < 0.05, **P < 0.01, and ***P< 0.001; n.s., non-significant difference, calculated using unpaired Student's *t* tests.

UDP-GlcNAc, in *ppg1Δ* cells (Fig 2B). These data reveal a higher gluconeogenic flux, coupled with increased carbon allocation towards synthesis of storage carbohydrates and cell wall precursors in *ppg1Δ* cells. This metabolic characterization of *ppg1Δ* cells thus revealed a specific imbalance in carbon allocation in the post-diauxic phase.

To understand if the Ppg1-dependent metabolic regulation is specific to the post-diauxic phase, we also assessed its role in the glycolytic phase of growth in glucose-replete conditions. For this, we compared these same metabolites at different time points starting from a glucose-replete medium. In *ppg1Δ* cells, the steady-state amounts of intermediates of glycolysis, precursors of cell wall and storage carbohydrates were unaltered compared to WT cells in the pre-diauxic phase. In *ppg1Δ* cells, the increased accumulation of these metabolites was observed only after glucose depletion in the post-diauxic phase (Fig 2C). We also assessed the effect of Ppg1 deletion on fermentation by measuring ethanol secretion in WT and *ppg1Δ* cells. In these conditions (prior to the diauxic shift), the amount of ethanol produced by *ppg1Δ* cells was indistinguishable from WT cells (Fig 2D). Collectively, these data suggest that the loss of Ppg1 does not affect glycolytic outputs in high glucose.

Since the carbon allocation towards synthesis of UDP-GlcNAc increases in *ppg1Δ* cells, we measured the levels of chitin from the cell walls of post-diauxic cells. Consistently, chitin levels increased in *ppg1Δ* cells (S2B Fig). Increased chitin accumulation is known to sensitize cells to cell wall stress [47,48]; hence, we studied the growth of *ppg1Δ* cells in the presence of two cell wall stress agents, Congo red and Calcofluor white. Expectedly, (and as observed earlier [49]) the growth of *ppg1Δ* cells was reduced in the presence of either Congo red or Calcofluor white (S2C and S2D Fig).

Finally, we asked if Ppg1 regulates the expression of transcripts of enzymes involved in gluconeogenesis, storage carbohydrate metabolism and cell wall synthesis proteins to modulate gluconeogenic outputs. For this, we measured the transcript levels of these enzymes from post-diauxic WT and *ppg1Δ* cells. Notably, the transcript levels of these enzymes largely remain unchanged in *ppg1Δ* cells (Fig 2E). This data reveals that Ppg1-mediated carbon flux regulation does not involve transcript level changes. This further suggests that Ppg1 regulates gluconeogenic flux via mechanisms that involve some combination of allosteric, post-translational or mass action-based regulation.

Collectively, these results establish that Ppg1 phosphatase controls the allocation of carbon towards gluconeogenic outputs in the post-diauxic phase of growth.

## Ppg1 interacts with Far11 and regulates Far complex assembly

How might Ppg1 regulate this balance of carbon flux in glucose-depleted environments? Ppg1 is a PP2A-like phosphatase. Several PP2A phosphatases themselves lack substrate specificity, and therefore function with associated proteins that facilitate the transient interaction of the

phosphatase with its substrates [50,51]. We hypothesized that identifying interacting partners of Ppg1 could help us understand its specific mechanisms of regulation. To identify proteins interacting with Ppg1, we introduced a FLAG epitope tag at C-terminus of Ppg1 at its chromosomal locus, performed a FLAG affinity purification/elution from post-diauxic cells, with wild-type (untagged) cells as control. Immunoprecipitated fractions were resolved on SDS gels and were visualized by silver staining. The protein gel revealed obvious differences in the immunoprecipitated proteins between Ppg1 and control samples (Fig 3A). The gel fragments were excised, and proteins were identified using mass spectrometry. A total of 33 and 24 proteins were identified in Ppg1 immunoprecipitation fractions (replicate 1 and 2 respectively) (S1 Table). The list of proteins identified was filtered manually by eliminating contaminant proteins and proteins found in control fractions (non-specific interactors), and retaining only those present in both biological replicates. The proteins uniquely identified in the Ppg1 immunoprecipitation fraction as Ppg1 interactors included Tpd3, Far11, and Ssa1 (Fig 3A). Tpd3 is a regulatory sub-unit of the PP2A phosphatase and is therefore expected, while Ssa1 is a chaperone of Hsp70 family. Hence, these two interactions were not further considered.

The identification of Far11 protein in this context of metabolic adaptation was unexpected, since there is no reported role for this protein in metabolic homeostasis. Far11 is a component of the Far complex—a multiprotein scaffolding complex consisting of six subunits (Far11, Far8, Far3, Far7, Far9, and Far10 subunits) [52] (Fig 3B), where the interaction between Far8 and Far11 completes the complex assembly [53]. Currently, this complex is known to prevent mitophagy during nitrogen starvation, and also functions in pheromone-mediated cell cycle arrest and TORC2 signaling [54]. Previous studies have found that Ppg1 phosphatase can interact with the Far complex [44]. However, none of these roles are related to the homeostatic regulation of carbon metabolism.

Therefore, in order to better understand interactions or relationships between Far complex and Ppg1, we first biochemically characterized the interaction between the Far complex and Ppg1 phosphatase using co-immunoprecipitation. Specifically, we immunopurified Far11 from cells growing in post-diauxic phase and assessed the association of Ppg1 with immunopurified Far11 (Fig 3C). The immunopurified Far11 specifically co-precipitated Ppg1 in this experiment (Fig 3C). We next asked if the phosphatase catalytic activity of Ppg1 was required for Far complex assembly, and performed similar co-immunopurification experiments using WT and Ppg1$^{H111N}$ cells. Consistently, we observed that the co-immunoprecipitation of Far11 with Far8 is notably reduced in Ppg1$^{H111N}$ cells (Fig 3D). This suggests that the interaction between Far11 and Far8 depends on the phosphatase activity of Ppg1 (Fig 3D). These data are also consistent with a report suggesting that Ppg1 is required for Far complex assembly [45].

Since Ppg1 activity was critical for interaction between components of the Far complex, we asked if Far11 or Far8 might be dephosphorylated, in a Ppg1-dependent manner. To assess this, we investigated changes in the electrophoretic mobility of Far8 and Far11 on SDS-PAGE gels. Changes in the electrophoretic mobility of a protein are a well-established read-out of protein phosphorylation/dephosphorylation [55]. We compared the electrophoretic mobility of Far8 and Far11 in WT and Ppg1$^{H111N}$ on SDS-PAGE gels. While the mobility of Far8 was not altered in Ppg1$^{H111N}$ cells, Far11 mobility in Ppg1$^{H111N}$ cells was reduced compared to WT (Figs 3E and S3A). This would be consistent with Ppg1-dependent dephosphorylation of Far11, and therefore increased phosphorylation of Far11 in Ppg1$^{H111N}$ cells. To further examine this possibility, we treated protein extracts with alkaline phosphatase and monitored Far11 electrophoretic mobility. The phosphatase treatment of protein extracts from post-diauxic wild-type cells resulted in reduced Far11 mobility (Fig 3E). This suggests that Far11 is phosphorylated in post-diauxic WT cells (Fig 3E). Additionally, the phosphatase-treated Far11 from Ppg1$^{H111N}$ cells show further reduced electrophoretic mobility compared to

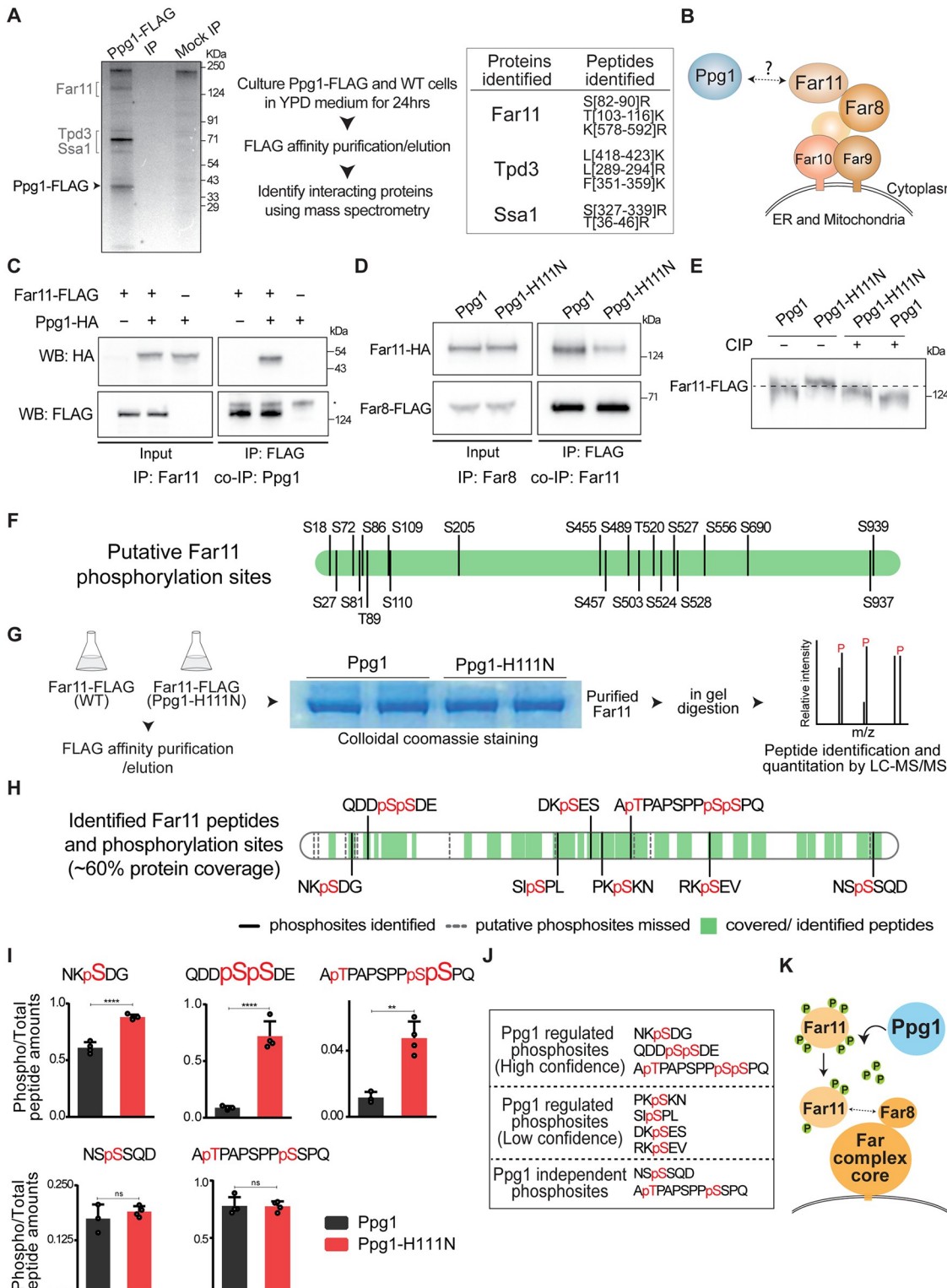

**Fig 3. Ppg1 interacts with Far11 to regulate its phosphorylation and Far complex assembly.** A) Identification of interacting partners of Ppg1. Lysates of WT and Ppg1-FLAG cells were subjected to FLAG affinity purification, immunopurified fractions were separated on SDS-PAGE followed by silver staining, and the immunopurified fractions were analyzed by mass spectrometry. The peptides identified for the proteins specifically interacting with Ppg1 are listed. The experiments were performed using 2 biological replicates. B) A schematic describing the composition of Far complex. C) Ppg1 interacts with Far11. FLAG-tagged Far11 was

immunoprecipitated from cells after 24hrs of growth in YPD medium and co-immunopurified HA-tagged Ppg1 was detected using western blotting. * denotes a non-specific binding of the antibody. A representative image is shown (n = 2). D) Requirement of Ppg1 activity for the interaction between Far11 and Far8. WT and Ppg1$^{H111N}$ cells containing HA-tagged Far11 and FLAG-tagged Far8 were cultured in YPD medium for 24hrs. Far8-FLAG was immunoprecipitated from these cells, and co-immunopurified Far11-HA was detected. The proteins were resolved using 4–12% gradient gels. A representative image is shown (n = 2). E) Regulation of Far11 post-translational modifications by Ppg1. WT and Ppg1$^{H111N}$ cells containing endogenously tagged Far11 with 3xFLAG epitope were cultured in YPD medium for 24hrs. After precipitating total protein, the electrophoretic mobility of Far11 was monitored on a 7% SDS-PAGE gel. For the phosphatase treatment, protein precipitates were dissolved in lysis buffer and treated with calf-intestinal phosphatase (CIP). A representative image is shown (n = 3). Also see S3A Fig. F) A schematic describing ser/thr residues on Far11 that are putative kinase/phosphatase sites. G) Purification of Far11 and identification of Far11 phosphosites. WT and Ppg1$^{H111N}$ cells containing endogenously tagged Far11 with 3xFLAG epitope were cultured in YPD medium for 24hrs. The Far11 was immunoprecipitated, samples were run on SDS-PAGE gel, bands corresponding to Far11 were excised and subjected to in-gel digestion. The Far11 phosphosites were identified by LC-MS/MS analysis. All experiments were performed using 4 biological replicates, and two replicates each of Far11 purifications from WT and Ppg1$^{H111N}$ cells are shown. H) Phosphorylated ser/thr residues on Far11. The peptide identification by mass spectrometry covered ~60% of the entire Far11 protein sequence. Identified peptide regions of Far11 are labeled in green, and the sequences of identified phosphosites are shown. Putative phosphosites that were missed in our analysis, in regions not detected by LC-MS/MS are marked using grey lines. Also see S3B Fig. I) Quantification of relative amounts of phosphorylated ser/thr peptides from Far11 in WT and Ppg1$^{H111N}$ cells. The relative amounts of phosphorylated peptides were calculated by comparing the abundance of phosphorylated peptides to the total peptide abundance. Data represented as a mean ± SD (n = 3). *P < 0.05, **P < 0.01, and ***P< 0.001; n.s., non-significant difference, calculated using unpaired Student's *t* tests. J) Classification of Far11 phosphosites based on their regulation by Ppg1 into three groups. The first group includes high-confidence sites, demonstrating a consistent change in phosphopeptide abundance across all 4 replicates of Ppg1$^{H111N}$ cells. The second group consists of low-confidence hits, including phosphopeptides with changes in abundance observed in only a few biological replicates of Ppg1$^{H111N}$ cells. The third group includes phosphopeptides that do not show any relative difference in Ppg1$^{H111N}$ cells. Also see S3B Fig. K) Schematic describing Ppg1 regulating phosphorylation of Far11 to regulate Far complex assembly.

phosphatase-treated Far11 from WT cells. Collectively, these data suggest that Far11 is post-translationally modified, and this modification depends on Ppg1 phosphatase activity. At this stage, while these experiments are consistent with a role of dephosphorylation in Far11 function and the assembly of the Far complex, these do not preclude other post-translational modifications in addition to phosphorylation.

## Ppg1 regulates the phosphorylation status of multiple ser/thr residues in Far11

A closer inspection of the Far11 sequence identifies over 19 ser/thr residues in Far11 that may be putatively phosphorylated (Fig 3F), based on prediction algorithms [56]. We therefore directly addressed the identification of Far11 ser/thr residues that are regulated by Ppg1. For this, we first immunopurified sufficient amounts of Far11 from post-diauxic WT and Ppg1$^{H111N}$ cells (Fig 3G). Subsequently, immunopurified fractions were resolved on SDS-PAGE gels, the bands corresponding to Far11 were excised, and in-gel digestion was carried out. The phosphorylated residues of Far11 were identified using LC-MS/MS. This LC-MS/MS based analysis was able to identify peptides that collectively covered ~60% of Far11 protein sequence. Notably, this analysis unambiguously identified ~8–12 phosphorylated ser/thr sites on Far11, across four independent replicates (respectively of WT and Ppg1$^{H111N}$ samples) (Fig 3H). From this analysis, we observed changes in the relative phosphorylation status of multiple serine residues in Ppg1$^{H111N}$ cells, while some residues did not show any changes in phosphorylation (Fig 3I).

We classified these phosphosites based on their regulation by Ppg1 into three groups. The first group includes high-confidence sites, which show a consistent increase in phosphopeptide abundance across all four replicates of Far11 isolated from Ppg1$^{H111N}$ cells (Fig 3J). This included residues pS81, pS109, pS110, and pS528, which all showed a significant increase in Ppg1$^{H111N}$ cells, suggesting increased phosphorylation of these serine residues (Figs 3I, 3J and

S3B). This would be consistent with reduced dephosphorylation of these residues in the catalytically inactive Ppg1$^{H111N}$ cells. The second group consisted of lower-confidence hits of phosphopeptides found in some biological replicates of Ppg1$^{H111N}$ cells. Notably, residues pS457, pS489, pS503, and pS690 showed an increase in some replicates of Ppg1$^{H111N}$, or were absent in WT samples (S3B Fig). This group of ser/thr residues are putatively regulated by Ppg1, but this identification is not unambiguous, due to technical limitations in mass spectrometry based detection. The third group included residues pT520, pS527, and pS939 which did not show any change in relative phosphorylation status in Ppg1$^{H111N}$ vs WT cells, indicating they are not under the regulation of Ppg1 (Figs 3I, 3J and S3B).

In summary, our analysis identified several, specific ser/thr residues in Far11 that undergo dephosphorylation in a Ppg1 dependent manner (Fig 3K). Collectively, our data now find that Ppg1 regulates the phosphorylation of multiple Far11 ser/thr residues, and this dephosphorylation of Far11 (in a Ppg1 dependent manner) is necessary for the assembly of the Far complex.

## Ppg1 regulates gluconeogenic carbon allocations via the Far complex assembly

Given these findings, we next asked if this Ppg1-dependent assembly of Far complex was required to regulate gluconeogenic outputs (Fig 4A). We compared levels of gluconeogenic intermediates, precursors of storage carbohydrates and cell wall in the post-diauxic phase from *far8Δ* and *far11Δ* cells. Similar to *ppg1Δ* cells, the amounts of all these metabolites increased in *far8Δ*, *far11Δ* and *far9Δ* cells (Figs 4B and S4A). Furthermore, the deletion mutants of components of Far complex were sensitive towards Congo red (S4B Fig), phenocopying the *ppg1Δ* cells. Collectively, these data find that the loss of the Far complex phenocopies the loss of Ppg1, and the assembled Far complex is required to appropriately apportion carbon allocations in post-diauxic cells.

To determine if Ppg1 and Far complex had independent roles in regulating carbon metabolism, or functioned sequentially within the same pathway, we generated double deletion mutants of Ppg1 and Far11 (*ppg1Δfar11Δ*). We assessed trehalose accumulation in *ppg1Δfar11Δ* cells, and found no additional increase in trehalose levels (Fig 4C). These cells phenocopied *ppg1Δ* or *far11Δ* cells. We also assessed the growth of the *ppg1Δfar11Δ* cells in the presence of Congo red, and observed no additive growth defects (Fig 4D). Overall, these findings strongly indicate that Ppg1 and Far complex function sequentially within the same pathway, with Ppg1 upstream of Far.

Summarizing, these data show that the Ppg1 phosphatase activity regulates the assembly of Far complex, which thereby regulates carbon flux towards gluconeogenic outputs important in post-diauxic cells (Fig 4E).

## Far complex tethering and not specific localization is required for gluconeogenic regulation

Interestingly, the Far complex is present in two sub-cellular localizations within a cell. One subpopulation localizes to the outer membrane of ER, and other to the mitochondrial outer membrane [45,57], both functioning as cytosol-facing complexes. We wondered if any one of the subpopulations of the Far complex was required for this regulation of carbon flux (Fig 5A). Therefore, we decided to study the role of each localized subpopulation of the Far complex in the context of this function.

The Far9 and Far10 proteins contain tail-anchor (TA) domains required to tether the complex to the ER or mitochondrial outer membrane. To determine which of the ER or the mitochondria-localized Far regulates post-diauxic carbon flux, we genome-engineered strains with

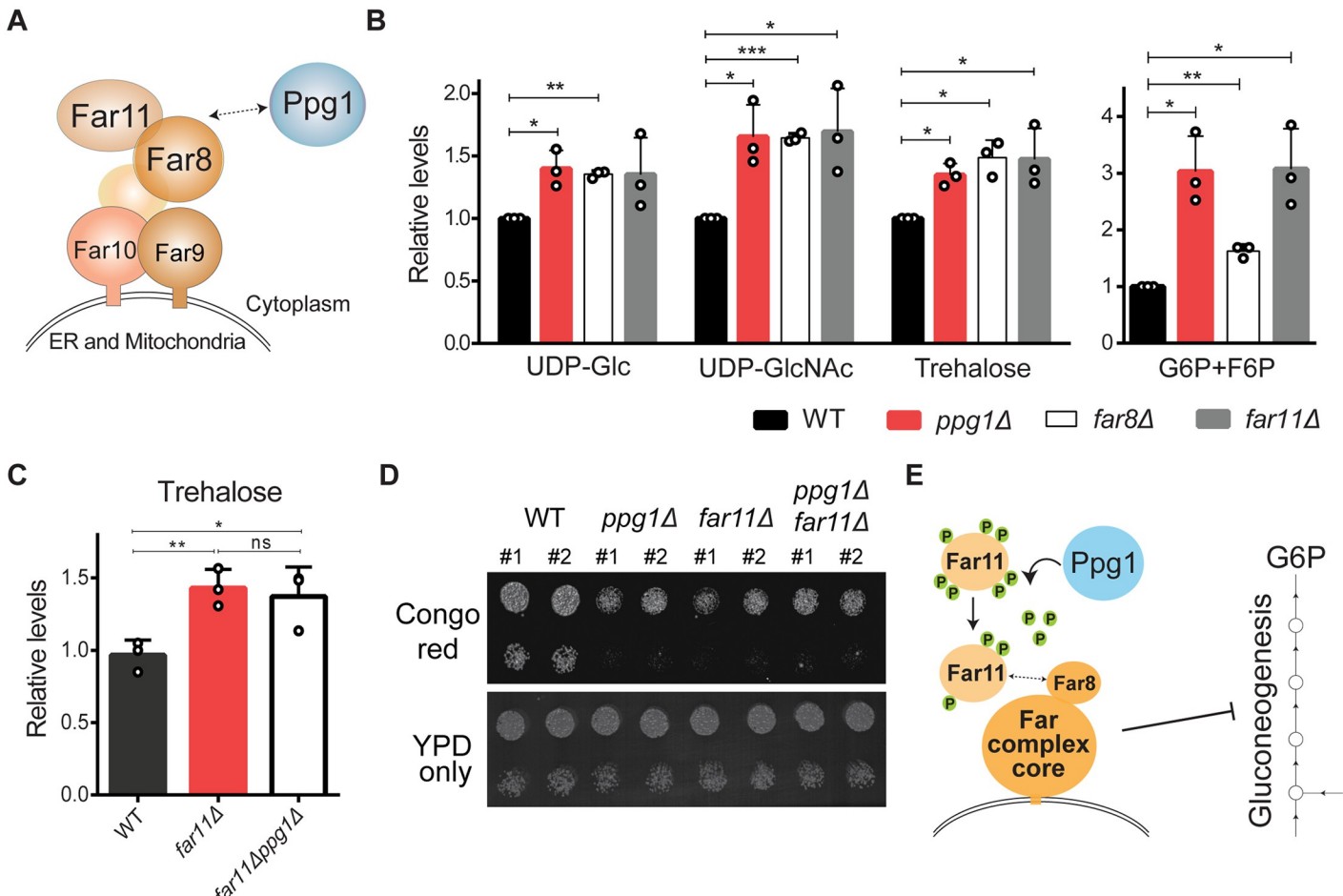

**Fig 4. Ppg1 regulates gluconeogenic carbon allocations via the Far complex.** A) A schematic describing the association between Ppg1 and Far complex. B) Relative steady-state amounts of specific gluconeogenic outputs from WT, *ppg1Δ*, *far8Δ*, and *far11Δ* cells after 24hrs of growth in YPD medium. Data represented as a mean ± SD (n = 3). *P < 0.05, **P < 0.01, and ***P< 0.001; n.s., non-significant difference, calculated using paired *t* tests. Also see S4A and S4B Fig. C) Relative trehalose levels in WT, *far11Δ*, and *ppg1Δfar11Δ* cells after 24hrs of growth in YPD medium. Data represented as a mean ± SD (n = 3). *P < 0.05, **P < 0.01, and ***P< 0.001; n.s., non-significant difference, calculated using unpaired Student's *t* tests. D) Growth of WT, *ppg1Δ*, *far11Δ*, and *ppg1Δfar11Δ* cells in the presence of Congo red. A serial dilution growth assay was carried out in the presence of Congo red. Congo red was used at a final concentration of 400 μg/ml. The images were taken after 60hrs of growth. A representative image is shown (n = 3). E) A schematic describing Ppg1 regulating Far complex assembly to control gluconeogenic carbon flux.

distinct Far localizations to either the ER or the mitochondrial outer membrane. For these strains, we replaced the native TA domain of Far9 with the TA domains of either Tom5 or Cyb5, which will respectively localize the Far complex specifically to only the mitochondrial surface or only the ER surface. The Far8-mNeonGreen from Far9-Cyb5 (ER-Far) and Far9-Tom5 (Mito-Far) cells showed exclusive ER or mitochondrial localization, respectively (Fig 5B). As readouts of wild-type Far complex function, we measured trehalose accumulation in post-diauxic phase from these mutant cells. Surprisingly, both the Far9-Cyb5 (ER-Far) as well as the Far9-Tom5 (Mito-Far) strains retained trehalose accumulation, similar to WT cells (Fig 5C), and distinct from the *far9Δ* or *ppg1Δ* cells. This suggested that carbon allocations remained unaltered in these cells with mitochondrial or ER surface localized Far complex (Fig 5C). Furthermore, these cells were not sensitive towards Congo red (Figs 5D and S5A), indicating that the cell wall composition of Far9-Cyb5 and Far9-Tom5 cells remained unaltered.

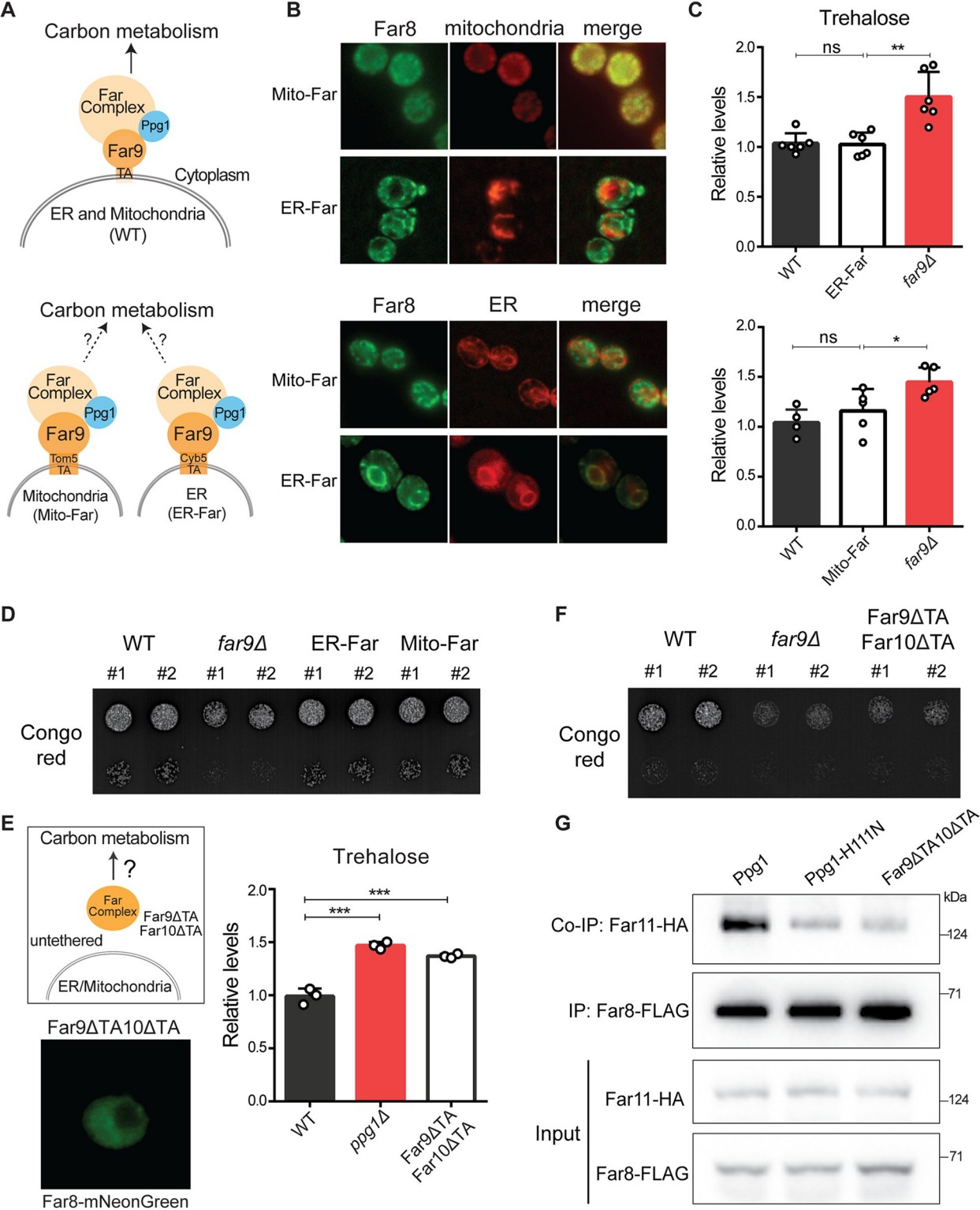

**Fig 5. Far complex tethering and not specific subcellular localization enables appropriate carbon allocations.** A) Schematic describing two sub-populations of Far complex and the possibility of a specific subpopulation involved in regulating carbon metabolism. The two sub-populations of Far complex are present at mitochondrial and ER outer membranes. The strains Mito-Far and ER-Far were constructed, and in these strains Far complex localizes specifically to mitochondria and ER respectively. B) The Mito-Far and ER-Far strains show distinct mitochondrial and ER localization of the Far complex. The Mito-Far and ER-Far cells were grown in YPD medium for 24hrs and analyzed by fluorescence microscopy.

Far8-mNeonGreen was used to visualize Far complex. Sec63-mCherry was used to visualize ER. Mitochondria were visualized using MitoTracker Red CMXRos. Representative images are shown. Also, see S5A Fig for control growth experiments for the distinct strains. Also, see S5B Fig. C) Effect of targeting of Far complex to ER and mitochondria on trehalose accumulation. WT, Mito-Far, ER-Far, and *far9Δ* cells were cultured in YPD medium for 24hrs and trehalose accumulation was measured from these cells. Data represented as a mean ± SD (n = 6 for ER-Far and n = 5 for Mito-Far). *P < 0.05, **P < 0.01, and ***P< 0.001; n.s., non-significant difference, calculated using unpaired Student's *t* tests. D) The growth of Mito-Far and ER-Far cells in presence of Congo red. A serial dilution growth assay was carried out in presence of Congo red using WT, Mito-Far, ER-Far, and *far9Δ* cells. Congo red was used at a final concentration of 400 μg/ml. The images were taken after 60hrs of growth. A representative image is shown (n = 2). Also, see S5A Fig for the growth of this strain in YPD. Also, see S5C Fig. E) Role of membrane tethering of Far complex on trehalose accumulation. The TA domains of Far9 and Far10 were deleted to disrupt the membrane tethering of Far complex (Far9ΔTA10ΔTA cells). WT, *far9Δ*, and Far9ΔTA10ΔTA cells were cultured in YPD medium for 24hrs and trehalose accumulation was measured from these cells. Data represented as a mean ± SD (n = 3). *P < 0.05, **P < 0.01, and ***P< 0.001; n.s., non-significant difference, calculated using unpaired Student's *t* tests. Also, see S5D and S5E Fig for localization of Far complex in Far9ΔTA and trehalose accumulation in this strain. See S5F Fig for glycogen amounts in Far9ΔTA10ΔTA strain. F) The growth of Far9ΔTA10ΔTA cells is attenuated in the presence of Congo red. A serial dilution growth assay was carried out in the presence of Congo red using WT, *far9Δ*, and Far9ΔTA10ΔTA cells. Congo red was used at a final concentration of 400 μg/ml. The images were taken after 48hrs of growth. A representative image is shown (n = 2). Also, see S5G Fig for the growth of this strain in YPD. G) The interaction between Far8 and Far11 in Far9ΔTA10ΔTA cells. WT, Ppg1$^{H111N}$, and Far9ΔTA10ΔTA cells expressing HA-tagged Far11 and FLAG-tagged Far8 were cultured in YPD medium for 24hrs. Far8-FLAG was immunoprecipitated from these cells, and co-immunoprecipitated Far11-HA was detected. A representative image is shown (n = 3).

Additionally, in the Mito-Far and ER-Far cells, the Far10 protein (which has an intact TA domain) can localize to the surface of both organelles. To ascertain whether the Far complex in Far9-Tom5 and Far9-Cyb5 strains is exclusively localized to the mitochondria or endoplasmic reticulum (respectively), we deleted the tail anchor domain of Far10 in these genetic backgrounds. We generated Far9-Cyb5/Far10ΔTA and Far9-Tom5/Far10ΔTA strains, and assessed their subcellular localization. Far9-Cyb5/Far10ΔTA and Far9-Tom5/Far10ΔTA cells showed exclusive ER or mitochondrial localization, respectively (S5B Fig). To study the Far complex function, we assessed growth of these cells in presence of Congo red. Both these strains phenocopied wild-type cells and were not sensitive towards Congo red (S5C Fig). These data therefore collectively suggest that the specific localization of the Far complex to both the ER and mitochondrial surface was not critical to control this function of regulating carbon metabolism.

Therefore, we hypothesized that perhaps the Far complex required membrane tethering on a suitable cytosol-facing surface for its complete assembly. In such a scenario, the tethering of the complex on an available cytosol-facing membrane surface, along with the regulated dephosphorylation by Ppg1, would determine the complete Far complex assembly, and thereby regulate carbon metabolism. To disrupt the membrane tethering of Far complex, we first removed the TA domain of only Far9. Surprisingly, these Far9ΔTA cells retained a membrane-localized Far8 protein (S5D Fig) and showed similar trehalose accumulation as WT cells (S5E Fig). Therefore, the deletion of the TA domain of Far9 alone is insufficient to disrupt the membrane tethering of the Far complex. We next removed the TA domains of both Far9 and Far10 within a single strain. The cells with this TA domain deletion in both Far9 and Far10 (Far9ΔTA10ΔTA) now showed cytosolic localization of Far8-mNeonGreen, and the membrane localization of Far complex was completely abolished (Fig 5E). We assessed trehalose accumulation and sensitivity to Congo red in these strains. Notably, the Far9ΔTA10ΔTA cells accumulated trehalose and glycogen in the post-diauxic phase, (Figs 5E and S5F), and were sensitive to Congo red (Figs 5F and S5G), similar to *ppg1Δ* cells. Collectively, these data reveal that the loss of tethering to both the ER and mitochondrial membrane disrupted the Far complex function and that the membrane tethering of the Far complex is required for appropriate carbon flux regulation in post-diauxic cells.

The simplest explanation for this reduced Far complex function in Far9ΔTA10ΔTA cells is that the tethering of Far to an available cytosol-facing membrane surface is required for the complete assembly of the complex. To therefore test if the Far complex assembly is disrupted

after the loss of membrane tethering, we examined the interaction between Far11 and Far8 using co-immunoprecipitation. In Far9ΔTA10ΔTA cells, the interaction between Far11 and Far8 was greatly reduced compared to the wild-type cells (Fig 5G), indicating that membrane tethering of Far complex is required for complete complex assembly.

These data collectively show the requirement of membrane tethering for the efficient assembly of the Far complex, and this assembled complex regulates carbon allocations towards gluconeogenic outputs. The specific subcellular localization of the Far complex to either the ER or mitochondria is itself not important for controlling carbon allocation.

## Glucose depletion increases amounts of the Far complex

Since the Ppg1-Far complex mediated regulation of carbon metabolism is specific to the post-diauxic phase, how responsive were the Far complex or Ppg1 itself to glucose levels? For this, we first measured amounts of Ppg1 and the components of Far complex during the course of growth, in pre-diauxic and post-diauxic phase cells. The amounts of Ppg1 remained constant during the course of growth in YPD (Fig 6A). In contrast, the levels of Far11 increased after 24hrs of growth in YPD (Fig 6B). From these findings, it can be inferred that the amount of Far complex itself increases in post-diauxic phase. We further examined the interaction between Far8 and Far11 from cells in the pre-diauxic and post-diauxic phases. In both pre-diauxic phase and post-diauxic phase cells, the interaction between Far11 and Far8 was retained (S6A Fig). However, using the electrophoretic mobility-based assay shown earlier, we observed that Far11 shows an electrophoretic mobility shift after glucose depletion (after 12 and 24hrs of growth in YPD) (Fig 6C). This data suggests that Far11 gets post-translationally modified as glucose depletes. The increased amounts and post-translational regulation of Far11 were also observed in Ppg1$^{H111N}$ cells (Fig 6C), indicating an alternative Ppg1-independent regulation of the Far complex in response to glucose depletion.

Finally, if glucose availability regulated the amounts of the Far complex, one possibility would be that the addition of glucose to post-diauxic cells would reduce Far complex amounts. To test this, we grew cells in high glucose and after cells reached the post-diauxic phase, we added glucose and compared the levels of Far11 protein. Glucose addition to post-diauxic cells reduced the amounts of Far11 to that observed in pre-diauxic phase of growth (Fig 6D). Furthermore, the amounts of a distinct Far complex protein, Far8, also similarly decreased after adding glucose to post-diauxic cells (S6B Fig). Together, we infer that the activity and amounts of Ppg1 are constitutive, but the amounts of the Far proteins are glucose-responsive (Fig 6E). The increased amounts of this assembled Far complex therefore modulate gluconeogenic outputs in post-diauxic cells.

## Ppg1-Far complex mediated carbon flux regulation enables cells to adapt to glucose depletion

Having identified this role of the Ppg1 phosphatase and Far complex in modulating post-diauxic carbon allocations towards gluconeogenesis, we asked if this Ppg1-Far mediated regulation enables cells to better adapt to glucose depletion. To address this, we designed a competitive growth-fitness experiment using WT and *ppg1Δ* cells, as illustrated in Fig 7A. For this, we used WT and *ppg1Δ* cells constitutively expressing mCherry and mNeonGreen fluorophores respectively to quantify different cells (see methods for details). To specifically address if Ppg1 enabled competitive growth in glucose-depleting environments, fluorescently labelled WT and *ppg1Δ* cells were started in equal proportions and grown in a glucose-replete medium for 24hrs (i.e., post-diauxic phase) and then shifted to a glucose-replete medium, and this process of transfers was repeated daily, and the relative proportion of WT and *ppg1Δ* cells were

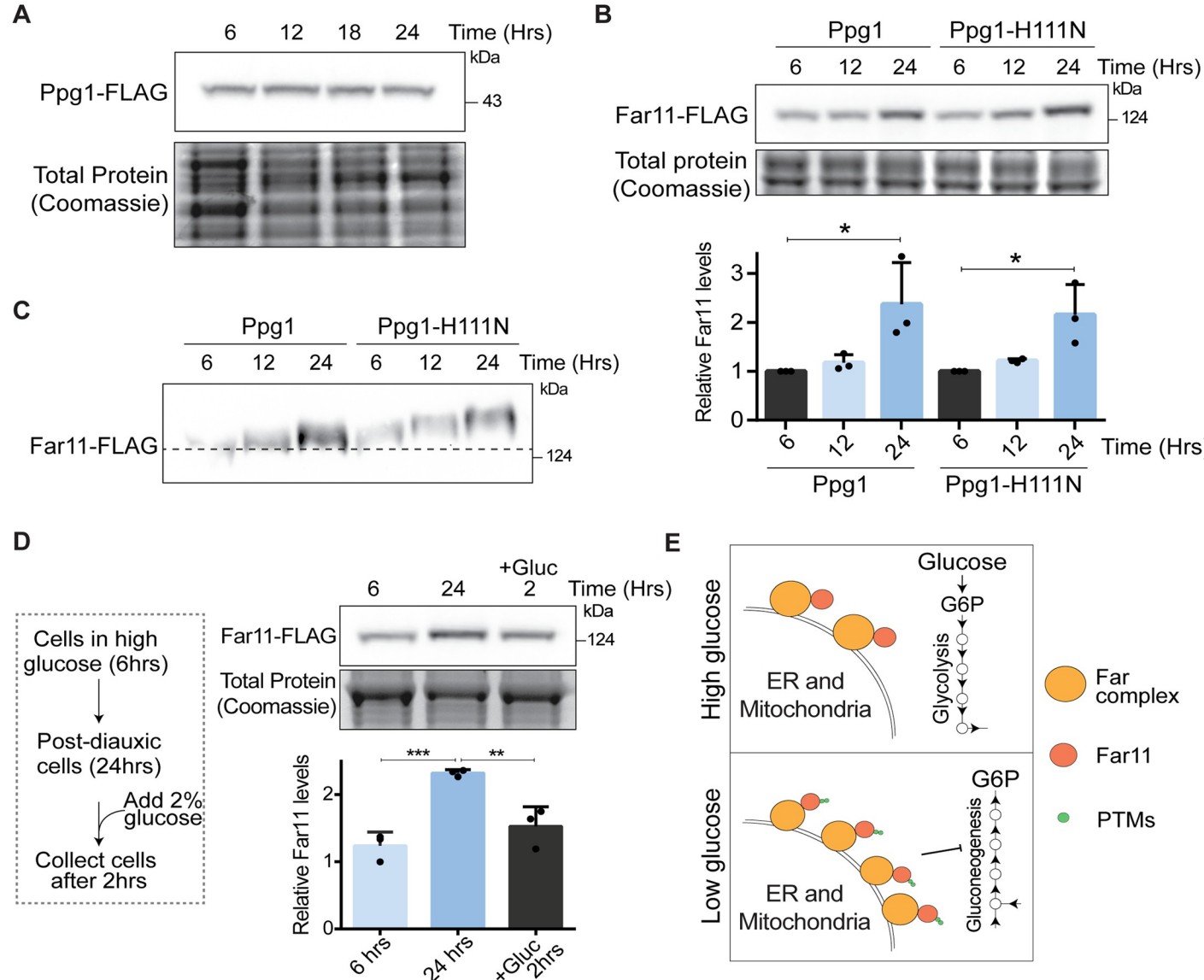

**Fig 6. Glucose depletion increases amounts of the Far complex.** A) Levels of Ppg1 protein during the course of growth. Cells with Ppg1 tagged with 3xFLAG epitope at the C terminus were cultured in YPD medium. Cells were collected at indicated time points and levels of Ppg1-FLAG were measured by western blotting. A portion of the gel was Coomassie stained and used as a loading control. A representative image is shown (n = 3). B) Levels of Far11 protein during the course of growth. The WT and Ppg1[H111N] cells containing endogenously tagged Far11 with 3xFLAG epitope were cultured in YPD medium. Cells were collected at indicated time points and levels of Far11-FLAG were measured by western blotting. A portion of the gel was Coomassie stained and used as a loading control. Western band quantification was done using ImageJ software. A representative image is shown (n = 3). *P < 0.05, **P < 0.01, and ***P< 0.001; n.s., non-significant difference, calculated using paired Student's *t* tests. Also, see S6A Fig for interactions between Far8 and Far11, and S6B Fig for Far8 protein amounts in different phases of growth. C) Far11 post-translational modifications during the course of growth. WT and Ppg1[H111N] cells containing endogenously tagged Far11 with 3xFLAG epitope were cultured in YPD medium. Cells were collected at indicated time points and Far11 mobility was monitored on a 7% SDS-PAGE gel. A representative image is shown (n = 3). D) Effect of glucose availability on amounts of Far11. Cells expressing FLAG-tagged Far11 were cultured in YPD medium. After 24hrs of growth, cells were diluted in the spent medium and 2% glucose was added. Cells were collected after 2hrs and levels of Far11 were measured by western blotting. A portion of the gel was Coomassie stained and used as a loading control. Western band quantification was done using ImageJ software. A representative image is shown (n = 3). *P < 0.05, **P < 0.01, and ***P< 0.001; n.s., non-significant difference, calculated using unpaired Student's *t* tests. Also, see S6B Fig. E) A schematic describing the effect of glucose availability on Far complex amounts.

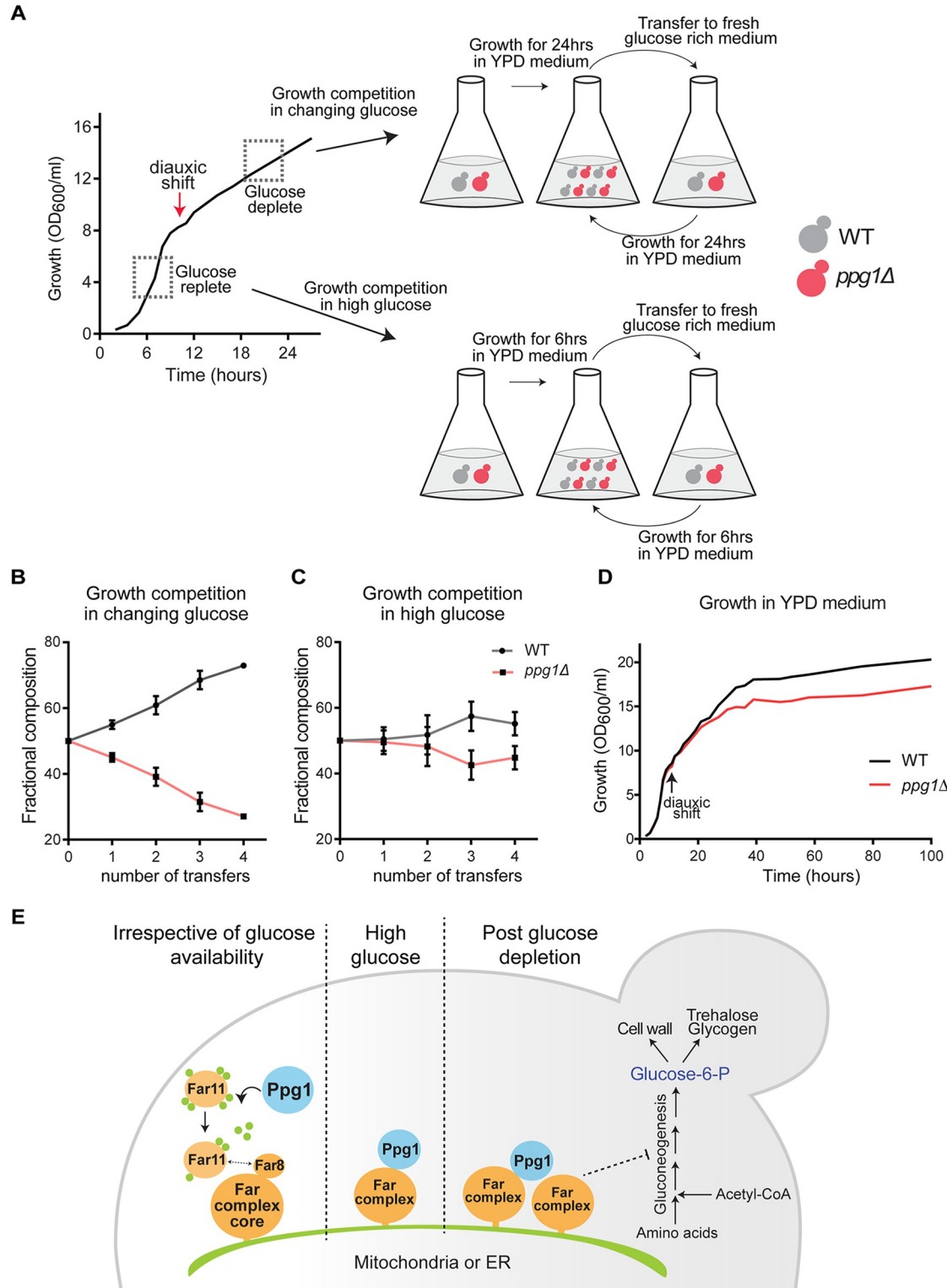

**Fig 7. Ppg1-Far complex mediated gluconeogenic flux regulation enables cells to adapt to glucose depletion.** A) A schematic describing the experimental details of growth competition experiment. WT cells expressing mCherry and *ppg1Δ* cells expressing mNeonGreen were mixed in equal proportion. These cells were grown in YPD medium and after growth for the indicated time were shifted to fresh YPD medium. The number of WT and *ppg1Δ* cells were calculated by fluorescence microscopy. B) Growth competition between WT and *ppg1Δ* cells in changing glucose conditions. The total culturing time for the competition experiment

was 96 hours. Data represented as a mean ± SD (n = 3). Also, see S7A Fig for a similar growth competition between WT and Far9ΔTA10ΔTA cells. C) Growth competition between WT and *ppg1Δ* cells in glucose-replete conditions. The total culturing time for the competition experiment was 24 hours. Data represented as a mean ± SD (n = 3). D) Comparative growth of WT and *ppg1Δ* cells in YPD medium. The cultures of WT and *ppg1Δ* were started at $OD_{600}$ of 0.2 in YPD medium and the growth was monitored. Also, see S7B Fig for a similar growth analysis of WT and *far11Δ* cells under the same conditions. E) Proposed model. Ppg1 phosphatase controls the assembly of the Far complex irrespective of glucose availability. Glucose availability regulates amounts of Far complex. Ppg1-Far complex controls carbon flux and gluconeogenic outputs specifically in the low glucose.

estimated (the experimental design is illustrated in Fig 7A). Notably, the fraction of WT cells substantially increased with each transfer (Fig 7B), suggesting that Ppg1-dependent regulation enables cells to adapt to glucose-depleting environments. In control experiments, competitive growth was compared in glucose-replete environments, where WT and *ppg1Δ* cells were grown in a glucose-replete medium, and cells were transferred to a fresh glucose-replete medium continuously every 6 hours before cells reached the diauxic phase (design in Fig 7A). Contrastingly, in this context, the relative proportion of WT and *ppg1Δ* cells remained similar after transfers, indicating that Ppg1 function did not affect growth fitness in high glucose (Fig 7C). To concurrently address the role of Far complex in enabling cells to adapt to glucose depletion, we carried out a similar competition experiment (as with *ppg1Δ*) with WT and Far9ΔTA10ΔTA cells. Note: the Far9ΔTA10ΔTA cells will not allow the Far complex to anchor and assemble within cells, as shown earlier, and therefore phenocopies *far9Δ*, and was utilized in this experiment for easier quantitative estimations based on fluorescence. Expectedly, the relative proportion of Far9ΔTA10ΔTA cells decreased during the course of the competition experiment (S7A Fig). We next examined the effect of loss of Ppg1 on steady-state batch culture growth, starting from a glucose-replete medium. The loss of Ppg1 did not affect growth in the glucose-replete log phase, but after cells entered the post-diauxic (glucose-depleted) phase, *ppg1Δ* cells showed reduced growth and a reduction in biomass accumulation (Fig 7D). Independently, we assessed the growth of *far11Δ* cells starting in glucose-replete conditions, and observed reduced growth of these cells specifically in the post-diauxic phase (S7B Fig), similar to *ppg1Δ* cells. Effectively, the loss of Ppg1 or the Far complex phenocopied each other, and collectively, these data reveal that Ppg1-Far mediated regulation enables adaptation and competitive growth fitness after glucose depletion.

Summarizing, we find that Ppg1 controls the assembly of the Far complex via the dephosphorylation of Far11. This increased Far complex after glucose depletion helps maintain appropriate allocations of carbon towards gluconeogenic, post-diauxic metabolism. This Ppg1-Far mediated regulation of carbon allocation thus allows cells to adapt and grow competitively post glucose depletion (Fig 7E).

## Discussion

In this study, we discover a role for the PP2A-like phosphatase Ppg1 in modulating carbon allocations, which enables cells to adapt and sustain competitive growth upon glucose depletion (Fig 7E). Ppg1 restrains the gluconeogenic carbon flux and allocation of carbon to gluconeogenic outputs as glucose depletes and cells enter the post-diauxic phase. Ppg1 mediates this function through an unexpected mechanism, where it controls the assembly of a large scaffolding protein complex, the Far complex (Fig 7E). The assembly of the Far complex requires the Ppg1 phosphatase activity, and the Ppg1-mediated Far complex assembly determines appropriate gluconeogenic outputs (Fig 7E).

There are other examples of signaling systems that regulate metabolic adaptation, which have typically focused on understanding the repression or activation of relevant transcriptional

outputs. For example, upon glucose depletion, the Snf1 kinase activates transcription factors such as Cat8 and Rds2, resulting in an increase in transcripts of key gluconeogenic enzymes [32,58,59]. In this context, phosphatases belonging to the PP2A family, particularly Pph21 and Pph22, regulate transcriptional outputs of glucose repressed genes [60,61]. Interestingly, and in contrast to these findings, the PP2A-like phosphatase Ppg1-mediated regulation identified in this study does not rely on changes in gene expression of glucose-repressed genes to control gluconeogenic flux (Fig 2E). Instead, this finding points towards regulation through other mechanisms that are driven by post-translational modifications, mass action, or enzyme concentration etc. This function of Ppg1, as uncovered in this study, differs from regulation mediated by related phosphatases. How might this occur? An underappreciated but important mediator of metabolic adaptation is the direct modulation of metabolic outputs or flux, through a combination of mass action and allosteric regulation (and without invoking transcriptional changes). Even in unicellular organisms like *S. cerevisiae*, over 50% of metabolic regulation occurs through such mechanisms [62]. In this study, the loss of Ppg1 increases the levels of gluconeogenic intermediates, precursors of cell wall and storage carbohydrates (Fig 2A). Increasing flux towards G6P and UDP-glucose would be one way of supporting the increased synthesis of storage carbohydrates without requiring alterations in enzyme levels, driven primarily by mass action. Classic studies of the trehalose synthesis enzymes in yeast [63,64] indicate this possibility. This increase in gluconeogenic flux in *ppg1Δ* cells indicates an imbalance in carbon allocations, resulting in increased consumption of amino acids towards gluconeogenic outputs. This therefore might limit their availability for other cellular processes. Although the gluconeogenic flux is higher in *ppg1Δ* cells, these cells have reduced growth in the post-diauxic phase. This plausible mode of regulation via Ppg1 could be systematically investigated in future studies, as an example of regulation mediated via some combination of mass action, concentration, allostery and enzyme regulation. These additional mechanisms (through scaffolding systems working together with signaling systems) to mediate overall metabolic outputs might be more prevalent than currently appreciated. In this context, we recently identified a signaling axis of the TORC1 scaffold Kog1 interacting with Snf1 (AMPK) to enable precise carbon allocations in nutrient-limited environments, ensuring optimum growth and adaptation during such conditions [32].

Notably, the Ppg1 phosphatase regulates post-diauxic carbon metabolism by modulating the assembly of the Far complex (Fig 3). Considering this requirement of Ppg1 to assemble this scaffolding complex and thereby constrain gluconeogenic flux, our study presents two intriguing possibilities: first, the Far complex scaffold could act as a facilitator, enabling interaction between Ppg1 and its other substrates (which regulate gluconeogenic outputs); and second, the primary function of Ppg1 is to facilitate Far complex assembly, which transiently brings to proximity other signaling proteins and enzymes that control gluconeogenesis. Both these possibilities (which are not mutually exclusive) merit detailed investigation. However, exploring these would require the development of new, proximity-based target identification systems for yeast that can identify transient protein-protein interactions. Separately, phospho-proteomics-based studies could provide avenues for identifying as yet unidentified substrates of Ppg1. However, phosphoproteomics approaches have been far more suited for elucidating kinase-mediated regulation, due to high substrate specificity of kinases [65]. Indeed, even in this study, using purified Far11, it was challenging to identify multiple Ppg1 activity dependent ser/thr phospho-sites on this protein. Identifying the specific substrates of phosphatases *in vivo* has posed significant challenges because of the nature of phosphatases like PP2A, which exhibit low substrate specificity and often have overlapping and compensatory outputs [66,67]. Hence, determining the specific outputs or substrates of phosphatases through these methods presents an especially formidable challenge, even in tractable systems such as yeast [68–70].

While the assembly of the Far complex is regulated by Ppg1 phosphatase activity, the Far complex amounts themselves increase upon depleting glucose. This collectively exemplifies the use of reversible phosphorylation to create a dynamic scaffolding complex assembly in order to modulate metabolic states. While scaffolding proteins can integrate various biological processes, the spectrum of regulatory possibilities they enable have not been fully explored [71]. Scaffolding assemblies can act as molecular facilitators to enable signaling proteins to transiently interact with correct partners, and also reduce non-specific interactions. For example, the presence of a scaffold in MAPK signaling increases the effective concentration of signaling components by nearly 3000-fold [72]. A scaffold could therefore modulate signal amplitude or intensity, both of which could be critical in metabolic adaptations. Such a mechanism allows cells to contextually tune outputs, without altering the signaling regulator (for example, the phosphatase) itself. A recent kinase-dependent example of such a scenario is where the yeast AMP-activated kinase (Snf1) phosphorylates Kog1, the scaffolding protein of TOR complex, leading to the disassembly of the TOR complex in extreme glucose starvation [73–75]. A parallel example is the contextual modulation of the activity of Snf1 by the TORC1 scaffold Kog1 in a TORC1 kinase-independent manner, to regulate carbon allocations as glucose depletes [32]. In the current study, the Ppg1 phosphatase functions to assemble a scaffolding complex, which could create localized pools of signaling that would regulate metabolic outputs, likely by facilitating context-dependent, specific interactions with substrates. This would be an additional way to contextually achieve mass action and/or allostery-based changes in metabolic flux, as discussed earlier. These are various extensions of the original conceptualization of 'metabolons' by Paul Srere [76,77]. These examples rely on various ways to form localized complexes, which themselves need not have enzymatic activity, but alter metabolic outputs. We can now imagine scenarios in fluctuating nutrient environments, where dynamic molecular assemblies can contextually tune signaling and metabolic outputs, to enable competitive growth.

In addition to this identified role for the Far complex in metabolic adaptation, this complex also participates in other processes such as TORC2 signaling, pheromone response, and mitophagy [44,52,54]. It is unclear how the Far complex might enable signal-specificity, and distinguish between different outputs, based on distinct inputs. Our study opens one possibility that the organelle-specific localization and post-translational regulation of Far complex could enable contextual regulation of outputs. Here, while the Far11 gets modified as glucose depletes and cells enter the post-diauxic phase, the proteome itself is distinct in composition as compared to a glucose-replete environment. The modifications that decorate Far11 specifically in post-diauxic cells might enable it to interact transiently with proteins that regulate post-diauxic metabolism, or assemble signaling hubs where a phosphatase or kinase could encounter a specific substrate. In order to understand how dynamically assembled scaffolds with varying localizations and modifications can regulate homeostatic outputs such as metabolic adaptations, we require new chemical biology approaches that stabilize low-affinity protein-protein interactions, or substrate-trapping mutants to identify transient substrates that are brought together by such signaling hubs [78]. This remains a key challenge in the context of protein phosphatases, which naturally interact with substrates with low affinities [79].

Finally, homologs of the Far complex have been found in diverse eukaryotes and are known as striatin-interacting phosphatase and kinase (STRIPAK) complexes [80]. The STRIPAK complexes regulate Hippo signaling, MAPK signaling, embryonic development, and so on [81–83]. Many STRIPAK components and Far complex components are evolutionarily conserved [80]. However, the kinases or phosphatases that associate with or regulate STRIPAK complexes are unclear, and it remains unknown as to how these complexes function as signaling hubs. Therefore, identifying regulators of the Far or STRIPAK complexes, and their

downstream effectors are exciting areas of future inquiry. Through the Far complex functions, it would be possible to uncover general mechanisms of how signaling hubs can assemble on available surfaces inside the cell, such as the ER or mitochondrial membrane surface. Multiple lines of inquiry now reveal that signaling pathways such as the TORC1 pathway can control contextual outputs from multiple, localized signaling hubs [84,85]. Such localized signaling hubs have not been explored in the context of homeostatic regulation of metabolic outputs. Our work identifying the Ppg1-mediated Far complex assembly exemplifies how a localized signaling hub could modulate metabolic outputs. This advances our basic understanding of how cells tune metabolism using localized signaling hubs, in order to adapt to changing nutrient environments. Such understanding might also allow us to optimize our use of cells as factories for metabolic engineering.

## Methods

### Yeast strains, media and growth conditions

The prototrophic CEN.PK, haploid "a" strain of *S.cerevisiae* was used in all the experiments. The strains with gene deletions and chromosomally tagged proteins were generated as described in [86]. For all the experiments, cells were cultured in YPD medium (1% yeast extract, 2% peptone, and 2% glucose) unless otherwise mentioned. For solid agar plates, YPD media was supplemented with 2% granular agar. For all the experiments, cells were grown overnight in YPD medium, and this primary culture was used to start a secondary culture at $OD_{600}$ of ~0.2. All the cultures were incubated at 30˚C/240 rpm. For the metabolic flux experiment, synthetic medium supplemented with amino acids (SCD—yeast nitrogen base without amino acids, all amino acids 2mM each with 2% glucose) was used.

### Serial dilution spot growth assay

For spot assay, ~1 $OD_{600}$ cells of respective strains were collected and washed with water. Serial dilutions were made in water and 5 μl of cell suspension was spotted on respective agar plates; growth was monitored for 48hrs. Congo red was used at a final concentration of 400 μg/ml.

### CRISPR based mutagenesis

Cells were first transformed with a plasmid expressing Cas9 (Addgene plasmid 43802) [87]. Guide RNAs (gRNAs) targeting genomic loci to be mutated were cloned in a plasmid with gRNA scaffold. The Cas9-expressing cells were transformed with gRNA plasmid and homology repair (HR) fragments. The cells were plated on YPD plates with selection drugs and the correct clones were confirmed using Sanger sequencing. The HR fragments used for transformation were synthesized.

### Western blot analysis

The cells were collected and protein extraction was carried out as described in [32]. Briefly, approximately ~10 $OD_{600}$ cells were collected by centrifugation and total protein was precipitated and extracted using 10% trichloroacetic acid and resuspended in SDS-glycerol buffer. The extracted proteins were estimated by BCA (bicinchoninic acid) protein assay kit. The samples were normalized for protein amounts and an equal amount of protein from all the samples were run on 4–12% bis-tris gels (Invitrogen, NP0336BOX). For the Far11 mobility shift assay, 7% SDS-PAGE gel was used. The relevant portion of gel was cut and used for transfer and blotting. The lower portion was stained with Coomassie blue for protein loading normalization. The blots were developed using the following antibodies: anti-FLAG raised in mouse (1:2000;

Sigma-Aldrich, F1804), anti-HA raised in mouse (1:2000; Sigma-Aldrich, 11583816001), anti-HA raised in rabbit (1:2000; Sigma-Aldrich, H6908), anti-mouse horseradish peroxidase (HRP)-conjugated antibody (1:4000; Cell Signaling Technology, 7076S), anti-rabbit HRP-conjugated antibody (1:4000; Cell Signaling Technology, 7074S). For chemiluminescence detection, western bright ECL HRP substrate (Advansta, K12045) was used. Alkaline phosphatase treatment was performed as described in [32,57].

## Immunoprecipitation and co-Immunoprecipitation

Immunoprecipitation was carried out as described in [32]. Cells (~50 $OD_{600}$) were collected by centrifugation, the pellet was flash-frozen in liquid nitrogen and stored at -80˚C. The pellet was resuspended in lysis buffer (50 mM HEPES buffer (pH 7.0), 50 mM NaF, 10% glycerol, 150 mM KCl, 1 mM EDTA, 2 mM sodium orthovanadate, 2 mM phenylmethylsulfonyl fluoride, 0.1 mM leupeptin, 2 mM pepstatin, and 0.25% Tween 20) and cell lysis was carried out by bead-beating. The supernatant was collected by centrifugation and was precleared by incubating with Dynabeads protein G beads (Invitrogen, 10004D). The beads were pulled down using DynaMag (Invitrogen, 12321D) and the precleared lysates were incubated with Dynabeads protein G beads conjugated with anti-FLAG antibody. The suspension was incubated for 2hrs at 4˚C with gentle rotation. The beads were pulled down using DynaMag and the immunoprecipitated proteins were eluted by incubating with FLAG peptide for 40 minutes. The immunoprecipitated proteins were detected by Western blotting using appropriate antibodies.

## Mass spectrometry analysis for proteomics

For the identification of interacting partners of Ppg1, cells expressing Ppg1 tagged with 3xFLAG epitope and untagged cells were cultured in YPD medium for 24hrs (post-diauxic phase). Cells were collected and lysed as mentioned above. The immunopurified fraction was run on SDS-PAGE gel, the gel pieces were cut and subjected to in-gel digestion. Specifically, the gel pieces were incubated with dithiothreitol followed by incubation with iodoacetamide. The alkylated proteins were digested using Trypsin (G biosciences) for 12hrs at 37 ˚C. Trypsin-digested peptides were dissolved in 0.1% formic acid and were analyzed on Thermo EASY-nLC 1200nano System coupled to a Thermo Scientific Orbitrap Fusion Tribrid Mass Spectrometer. The experiment was performed with 2 biological replicates. Data analysis was performed using Mascot distiller. Searches were conducted using 10 ppm peptide mass tolerance, product ion tolerance of 0.6 Da resulting in a 1% false discovery rate. Mascot score and emPAI values were used to compare protein abundance [88]. S1 Table provides the complete proteomics raw data for Ppg1 interactors. The mass spectrometry proteomics data have been deposited to the ProteomeXchange Consortium via the PRIDE [89] partner repository with the dataset identifier PXD049793.

For identifying Far11 phosphorylation sites, wild-type and Ppg1 catalytically inactive (Ppg1$^{H111N}$) cells expressing Far11 tagged with 3xFLAG tag were cultured in YPD medium for 24 hours. Approximately 750 $OD_{600}$ cells were collected and lysed as mentioned above. FLAG affinity immunopurification was carried out, samples were run on SDS-PAGE gel, and gels were stained with colloidal coomassie blue. The band corresponding to Far11 was excised and subjected to in-gel trypsin digestion. Approximately 5 μg of Far11 protein was purified and utilized for LC-MS/MS analysis. The experiment was performed with 4 biological replicates. Data analysis was performed using Proteome Discoverer [90]. Searches were conducted using 10 ppm peptide mass tolerance, product ion tolerance of 0.6 Da resulting in a 1% false discovery rate. The relative amounts of phosphorylated peptides were calculated by comparing the abundance of phosphorylated peptides to the total peptide abundance, for each sample. S2

Table provides the complete proteomics raw data. The mass spectrometry proteomics data have been deposited to the ProteomeXchange Consortium via the PRIDE [89] partner repository with the dataset identifier PXD049989.

## Trehalose and Glycogen measurements

Trehalose and glycogen measurements were carried out as described in detail in [91]. Briefly, 10 $OD_{600}$ cells were collected and lysed by incubating with 0.25M $Na_2CO_3$ at 98˚C for 4hrs. The pH of solution was adjusted to 5.2 by the addition of appropriate amounts of 1M acetic acid and 0.2M $CH_3COONa$. For trehalose estimation, the solution was incubated with trehalase (0.025 U/ml; Sigma-Aldrich, T8778) at 37˚C overnight with gentle rotation. For glycogen measurement, the solution was incubated with amyloglucosidase (1 U/ml; Sigma-Aldrich, 10115) at 57˚C overnight. The amount of glucose released was measured by glucose assay kit (Sigma-Aldrich, GAGO20). The experiment was performed with 3 biological replicates and observed values were plotted using GraphPad Prism. Statistical significance was calculated using unpaired Student's *t* test.

## Metabolite extraction and LC-MS/MS analysis

The metabolite extraction and analysis were carried out following protocols described in [92]. For each experiment, 10 $OD_{600}$ cells were used for metabolite extraction. First, the cells were quenched for 5 minutes in 60% methanol (maintained at -45˚C). After centrifugation, the cell pellet was resuspended in the extraction buffer (75% ethanol) and kept at 80˚C followed by incubation on ice and centrifugation. The supernatant was collected, dried, and then stored at -80˚C till further use. Metabolites were separated on Synergi 4-μm Fusion-RP 80 Å (150 × 4.6 mm) LC column (Phenomenex, 00F-4424-E0) using the Waters Acquity UPLC system. The solvents used for separating sugar phosphatases and trehalose are the following: 5 mM ammonium acetate in water (solvent A) and acetonitrile (solvent B). Solvents used for separating amino acids are 0.1% formic acid in water (solvent A) and 0.1% formic acid in methanol (solvent B). The metabolites were detected using ABSciex QTRAP 6500 mass spectrometer. The data was acquired using Analyst 1.6.2 software (Sciex) and analyzed using MultiQuant version 3.0.1 (Sciex). Analyzed data was plotted using GraphPad Prism. Raw data and LC-MS/MS parameters are in S3 and S4 Tables.

## $^{13}$C carbon flux measurements

Protocols for measuring $^{13}$C carbon flux, and specifically gluconeogenic flux, are extensively described in earlier studies [91,92]. For measuring carbon flux towards gluconeogenesis, cells were grown in SCD medium for 24hrs and then pulsed with 1% $^{13}C_2$-acetate (Cambridge Isotope Laboratories, CLM-440). Metabolites were extracted after 30 minutes of labelled acetate addition. Samples were analyzed as described earlier. The addition of intensity of all labelled intermediates detected for a metabolite was calculated as total $^{13}$C label incorporation. The total label incorporation was normalized with WT values to calculate relative $^{13}$C label incorporation. Statistical significance was calculated using paired Student's *t* test. A table with Q1 and Q3 parameters for all metabolites and their $^{13}$C labelled forms is provided (S4 Table). In addition, the raw values for all mass spectrometry measurements are provided in S3 Table.

## Ethanol estimation assay

The ethanol estimation was carried out as described in [93]. Briefly, cells were grown in YPD medium to an $OD_{600}$ of ~0.8, cells were centrifuged and 5ml of supernatant was collected in a

fresh tube. 1 ml of Tri-n-butyl phosphate (TBP) was added and vortexed for 5 minutes. After centrifugation top layer was transferred to a fresh tube and mixed with an equal volume of potassium dichromate. The mixture was incubated for 10 minutes and absorbance was measured at 595nm. Statistical significance was calculated using unpaired Student's *t* test.

### Fluorescence microscopy

Cells expressing fluorescently tagged proteins were cultured in YPD medium for 24hrs and then visualized using a microscope (Olympus BX53) with 100 X objective. To stain mitochondria, cells were incubated with MitoTracker Red CMXRos. Far8 was tagged with mNeonGreen for visualizing the Far complex. Sec63-mCherry was used to visualize ER. The experiments were repeated using 3 biological replicates.

### Competition growth-fitness assay

WT cells expressing mCherry and *ppg1Δ* cells expressing mNeonGreen from chromosomal loci were cultured separately in YPD medium. These cultures were mixed together at an $OD_{600}$ of 0.2 each in YPD medium. Cultures were grown for 24hrs and then subcultured at an $OD_{600}$ of 0.2 in fresh YPD medium. The cells were allowed to grow for 24hrs and were again shifted to a fresh YPD medium, and these transfers were repeated several times. The relative proportion of WT and *ppg1Δ* cells were estimated by counting the number of mCherry and mNeonGreen positive cells using fluorescence microscopy. The experiment was performed with 3 biological replicates and a minimum of 100 cells were counted for each replicate. As a control, fluorescently labelled WT and *ppg1Δ* cells were cultured together in YPD medium for 6hrs, and cells were transferred to a fresh YPD medium, and these transfers were repeated. The fractional composition of WT and *ppg1Δ* cells was calculated and plotted.

### RNA isolation and quantitative real-time PCR (qRT-PCR) analysis

WT and *ppg1Δ* cells were cultured in YPD medium for 24 hours. Cells were harvested by centrifugation, and RNA isolation was carried out using hot acid phenol method [94]. cDNA was synthesized with random primers and SuperScript II reverse transcriptase. Relative transcript quantifications were done by real-time PCR using Maxima SYBR Green master mix (Thermo Fisher). TAF10 was used as an internal normalization control. The experiments were repeated using 3 biological replicates. Statistical significance was calculated using unpaired Student's *t* test.

### Data visualization and statistical test

All the graphs were plotted and analyzed using GraphPad Prism 6. Two-tailed unpaired Student's *t* test was used to estimate statistical significance unless otherwise specified. For the flux experiment, the raw intensity values for each replicate were normalized to respective wild-type control and paired Student's *t* test was used to estimate statistical significance. For the mass spectrometry experiment examining metabolite levels in Far complex mutants, a one-tailed paired Student's *t* test was used to estimate statistical significance. P values and *n* for corresponding experiments have been specified in figure legends. Raw data numerical values are provided in S3 and S5 Tables.

## Supporting information

**S1 Fig.** A) A list of protein phosphatases in *S. cerevisiae* and the phosphatase mutants used in the study. Trehalose accumulation was measured after 24hrs of growth in YPD medium. The

mean trehalose accumulation was obtained from 2 biological replicates. B) Trehalose amounts in log and post-diauxic phase of growth. Trehalose levels were measured after 4hrs and 24hrs of growth in YPD medium. Data represented as a mean ± SD (n = 3). *P < 0.05, **P < 0.01, and ***P< 0.001; n.s., non-significant difference, calculated using unpaired Student's $t$ tests. C) Effect of deletion of Atg32 on trehalose accumulation in WT and $ppg1\Delta$ cells. Trehalose accumulation was measured after 24hrs of growth in YPD medium. Data represented as a mean ± SD (n = 3). *P < 0.05, **P < 0.01, and ***P< 0.001; n.s., non-significant difference, calculated using unpaired Student's $t$ tests. D) Relative trehalose levels in wild-type and Ppg1-FLAG cells after 24hrs of growth in YPD medium. Data represented as a mean ± SD (n = 3). *P < 0.05, **P < 0.01, and ***P< 0.001; n.s., non-significant difference, calculated using unpaired Student's $t$ tests. E) Effect of H111N point mutation on protein levels of Ppg1. The WT and Ppg1$^{H111N}$ cells containing endogenously tagged Ppg1 with 3xFLAG epitope were cultured in YPD medium. Cells were collected after 24hrs of growth and the levels of Ppg1 were measured by western blotting. A portion of the gel was Coomassie stained and used as a loading control. Western blot quantification was done using ImageJ software. A representative image is shown (n = 3). Quantification data represented as a mean ± SD (n = 3). *P < 0.05, **P < 0.01, and ***P< 0.001; n.s., non-significant difference, calculated using unpaired Student's $t$ tests.
(TIF)

**S2 Fig.** A) Relative steady-state amounts of specific gluconeogenic intermediates, precursors of cell wall and storage carbohydrates, and amino acids in WT, $ppg1\Delta$, and Ppg1$^{H111N}$ cells after 24hrs of growth in YPD medium. Data represented as a mean ± SD (n = 3). *P < 0.05, **P < 0.01, and ***P< 0.001; n.s., non-significant difference, calculated using unpaired Student's $t$ tests. B) Relative chitin levels in cell walls of WT and $ppg1\Delta$ cells after 24hrs of growth in YPD medium. Data represented as a mean ± SD (n = 3). *P < 0.05, **P < 0.01, and ***P< 0.001; n.s., non-significant difference, calculated using unpaired Student's $t$ tests. C) The growth of WT and $ppg1\Delta$ cells in presence of Congo red. A serial dilution growth assay was carried out in presence of Congo red using WT and $ppg1\Delta$ cells. Congo red was used at a final concentration of 400 µg/ml. The images were taken after 60hrs of growth. A representative image is shown (n = 3). D) The growth of WT and $ppg1\Delta$ cells in the presence of Calcofluor white. A serial dilution growth assay was carried out in the presence of Calcofluor white. Calcofluor white was used at a final concentration of 50 µg/ml. The images were taken after 60hrs of growth.
(TIF)

**S3 Fig.** A) Regulation of Far8 post-translational modifications by Ppg1. WT and Ppg1$^{H111N}$ cells containing endogenously tagged Far8 with 3xFLAG epitope were cultured in YPD medium for 24hrs. Far8 mobility was monitored on a 7% SDS-PAGE gel. B) Schematic describing the Far11 phosphosites identified in various replicates of both wild-type and Ppg1$^{H111N}$ cells. The presence of a specific phosphopeptide in an individual replicate is denoted by a green box.
(TIF)

**S4 Fig.** A) Relative trehalose levels in WT, $ppg1\Delta$, and $far9\Delta$ cells after 24hrs of growth in YPD medium. Data represented as a mean ± SD (n = 3). *P < 0.05, **P < 0.01, and ***P< 0.001; n.s., non-significant difference, calculated using unpaired Student's $t$ tests. B) The growth of $far8\Delta$, and $far11\Delta$ cells in presence of Congo red. A serial dilution growth assay was carried out in presence of Congo red using WT, $ppg1\Delta$, $far8\Delta$, and $far11\Delta$ cells. Congo red was used at a final concentration of 400 µg/ml. The images were taken after 60hrs of growth. A

representative image is shown (n = 3).
(TIF)

**S5 Fig.** A) The growth of Mito-Far and ER-Far cells in YPD medium. A serial dilution growth assay was carried out using WT, Mito-Far, ER-Far, and *far9Δ* cells. The images were taken after 24hrs of growth. A representative image is shown (n = 2). B) The Mito-Far Far10ΔTA and ER-Far Far10ΔTA strains show distinct mitochondrial and ER localization of the Far complex. The Mito-Far and ER-Far cells were grown in YPD medium for 24hrs and analyzed by fluorescence microscopy. Far8-mNeonGreen was used to visualize Far complex. Sec63-m-Cherry was used to visualize ER. Mitochondria were visualized using MitoTracker red CMXRos. C) Growth of Mito-Far Far10ΔTA and ER-Far Far10ΔTA cells in presence of Congo red. A serial dilution growth assay was carried out in presence of Congo red using WT, Mito-Far Far10ΔTA, ER-Far Far10ΔTA, and *ppg1Δ* cells. Congo red was used at a final concentration of 400 μg/ml. The images were taken after 60hrs of growth. A representative image is shown (n = 2). D) The subcellular localization of Far complex in Far9ΔTA cells. Far8-mNeon-Green was used to visualize the localization of Far complex in these cells. E) Relative trehalose levels in WT, *far9Δ*, and Far9ΔTA cells after 24hrs of growth in YPD medium. Data represented as a mean ± SD (n = 2). *$P < 0.05$, **$P < 0.01$, and ***$P < 0.001$; n.s., non-significant difference, calculated using unpaired Student's *t* tests. F) Relative glycogen levels in WT, *far9Δ*, and Far9ΔTA10ΔTA cells after 24hrs of growth in YPD medium. Data represented as a mean ± SD (n = 3). *$P < 0.05$, **$P < 0.01$, and ***$P < 0.001$; n.s., non-significant difference, calculated using unpaired Student's *t* tests. G) The growth of Far9ΔTA10ΔTA cells in YPD medium. A serial dilution growth assay was carried out using WT, *far9Δ*, and Far9ΔTA10ΔTA cells. The images were taken after 24hrs of growth. A representative image is shown (n = 2).
(TIF)

**S6 Fig.** A) Interaction between Far8 and Far11 in different phases of growth. WT cells containing HA-tagged Far11 and FLAG-tagged Far8 were cultured in YPD medium. Cells were collected after 6hrs and 24hrs. Far8-FLAG was immunoprecipitated from these cells, and co-immunopurified Far11-HA was detected. A representative image is shown (n = 3). B) Effect of glucose availability on amounts of Far8. Cells expressing FLAG-tagged Far8 were cultured in YPD medium. After 24hrs of growth cells were diluted in the spent medium and 2% glucose was added. Cells were collected after 2hrs and levels of Far8 were measured by western blotting. A portion of the gel was Coomassie stained and used as a loading control. Western band quantification was done using ImageJ software. A representative image is shown (n = 3). *$P < 0.05$, **$P < 0.01$, and ***$P < 0.001$; n.s., non-significant difference, calculated using unpaired Student's *t* tests.
(TIF)

**S7 Fig.** A) Competitive growth between WT and Far9ΔTA10ΔTA cells in changing glucose conditions. The total culturing time for the competition experiment is 96 hours. Data represented as a mean ± SD (n = 3). B) Comparative growth of WT and *far11Δ* cells in YPD medium. The cultures of WT and *far11Δ* were started at $OD_{600}$ of 0.2 in YPD medium and the growth was monitored.
(TIF)

**S1 Table. List of Ppg1 interacting proteins identified by mass spectrometry.**
(XLSX)

**S2 Table. List of Far11 peptides identified by mass spectrometry.**
(XLSX)

**S3 Table. Raw data for all the metabolomics experiments.**
(XLSX)

**S4 Table. Parent and daughter ion m/z parameters for reported metabolites.**
(XLSX)

**S5 Table. Raw data numerical values underlying Figs 1–7 and S1–S7.**
(XLSX)

## Acknowledgments

We acknowledge extensive use of the NCBS-inStem-CCAMP mass spectrometry facilities.

## Author Contributions

**Conceptualization:** Shreyas Niphadkar, Sunil Laxman.

**Data curation:** Shreyas Niphadkar, Lavanya Karinje, Sunil Laxman.

**Formal analysis:** Shreyas Niphadkar, Lavanya Karinje, Sunil Laxman.

**Investigation:** Shreyas Niphadkar, Lavanya Karinje, Sunil Laxman.

**Methodology:** Shreyas Niphadkar.

**Project administration:** Sunil Laxman.

**Supervision:** Sunil Laxman.

**Validation:** Shreyas Niphadkar, Lavanya Karinje.

**Visualization:** Shreyas Niphadkar, Lavanya Karinje, Sunil Laxman.

**Writing – original draft:** Shreyas Niphadkar, Sunil Laxman.

**Writing – review & editing:** Shreyas Niphadkar, Sunil Laxman.

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
