## [Decision Letter · Decision Letter 0]

3 Dec 2023

Dear Dr Laxman,

Thank you very much for submitting your Research Article entitled 'The PP2A-like phosphatase Ppg1 mediates assembly of the Far complex to balance gluconeogenic outputs and adapt to glucose depletion' to PLOS Genetics. The manuscript was fully evaluated at the editorial level and by independent peer reviewers, some of whom also reviewed the initial submission to Review Commons. We appreciate the substantial revisions that were made following the original comments. However, although two of the reviewers state that most points were addressed, one still has substantial concerns. I agree with this reviewer (Reviewer 2) that is necessary to show that Ppg1 directly dephosphorylates Far11 in order to support your hypotheses. I appreciate that detailed phosphoproteomics may be complex; however, if it is not possible, some other convincing experimental analysis should be performed.

Based on the reviews, we will not be able to accept this version of the manuscript, but we would be willing to review a much-revised version. We cannot, of course, promise publication at that time.

If you decide to revise the manuscript for further consideration at PLOS Genetics, please aim to resubmit within the next 60 days, unless it will take extra time to address the concerns of the reviewers, in which case we would appreciate an expected resubmission date by email to plosgenetics@plos.org.

We are sorry that we cannot be more positive about your manuscript at this stage. Please do not hesitate to contact us if you have any concerns or questions.

Yours sincerely,

Geraldine Butler

Section Editor

PLOS Genetics

Geraldine Butler

Section Editor

PLOS Genetics

Reviewer's Responses to Questions

**Comments to the Authors:**

Reviewer #1: The authors have addressed the points raised satisfactorily

Reviewer #2: In this revised manuscript, the authors have failed to adequately respond to this reviewer's first round comments. Most of the comments refer to the proposed experiments as future research and make no effort to clarify them.

The authors found ppg1-deleted strain to be the most defective in metabolic adaptation among the available phosphatase deletion strains. However, there are many factors other than phosphatase that play important roles in metabolic adaptation, and it is unclear whether ppg1 has a comparable role or is only slightly involved. Therefore, the value of this paper is to elucidate the mechanism of why Ppg1 affects metabolic adaptation. It has already been shown that Ppg1 and the Far complex are interdependent, but the authors have newly proposed that Ppg1 can form the Far complex by dephosphorylating Far11. This model needs to be fully demonstrated before publication.

Specific comments

1.Figure 3E: Why does the molecular weight of Far11-FLAG after CIP treatment differ between the Ppg1-H11N and Ppg1 strains? Both should be fully dephosphorylated; it is possible that modifications other than phosphorylation may have occurred by Ppg1.

2. Figure 4A-C: In the presence of Far10, some Far complexes can localize to both mitochondria and ER. Even if the amount is undetectable by fluorescence microscopy, the possibility that it affects the phenotype cannot be ruled out.

3. it is necessary to prove the model that Ppg1 dephosphorylates Far11 and the dephosphorylated Far11 forms Far complex. As shown in comment 4 of first round, identification of the phosphorylation site of Far11 would be the easiest way, but other methods are also acceptable.

Reviewer #3: The authors have successfully addressed the concerns raised previously.

**Have all data underlying the figures and results presented in the manuscript been provided?**

Reviewer #1: Yes

Reviewer #2: Yes

Reviewer #3: Yes

PLOS authors have the option to publish the peer review history of their article (what does this mean?). If published, this will include your full peer review and any attached files.

Reviewer #1: No

Reviewer #2: No

Reviewer #3: No

---

## [Editor Report · Decision Letter 1]

27 Feb 2024

Dear Dr Laxman,

We are pleased to inform you that your manuscript entitled "The PP2A-like phosphatase Ppg1 mediates assembly of the Far complex to balance gluconeogenic outputs and enables adaptation to glucose depletion." has been editorially accepted for publication in PLOS Genetics. Congratulations!

Yours sincerely,

Geraldine Butler

Section Editor

PLOS Genetics

Comments from the reviewers (if applicable):

**Data Deposition**

http://datadryad.org/submit?journalID=pgenetics&manu=PGENETICS-D-23-01216R1

**Press Queries**

---

## [Editor Report · Acceptance letter]

4 Mar 2024

PGENETICS-D-23-01216R1 

The PP2A-like phosphatase Ppg1 mediates assembly of the Far complex to balance gluconeogenic outputs and enables adaptation to glucose depletion. 

Dear Dr Laxman, 

We are pleased to inform you that your manuscript entitled "The PP2A-like phosphatase Ppg1 mediates assembly of the Far complex to balance gluconeogenic outputs and enables adaptation to glucose depletion." has been formally accepted for publication in PLOS Genetics! Your manuscript is now with our production department and you will be notified of the publication date in due course.

With kind regards,

Zsofia Freund

PLOS Genetics

On behalf of:
